# Learning Joint Interventional Effects from Single-Variable Interventions in Additive Models

**Armin Kekić** [1]    **Sergio Hernan Garrido Mejia** [1 2]    **Bernhard Schölkopf** [1 3 4]

## Abstract

Estimating causal effects of joint interventions on multiple variables is crucial in many domains, but obtaining data from such simultaneous interventions can be challenging. Our study explores how to learn joint interventional effects using only observational data and single-variable interventions. We present an identifiability result for this problem, showing that for a class of nonlinear additive outcome mechanisms, joint effects can be inferred without access to joint interventional data. We propose a practical estimator that decomposes the causal effect into confounded and unconfounded contributions for each intervention variable. Experiments on synthetic data demonstrate that our method achieves performance comparable to models trained directly on joint interventional data, outperforming a purely observational estimator.

## 1. Introduction

Understanding the effects of interventions is fundamental across many domains, from designing public health policies to optimizing business operations and administering medical treatments. Particularly challenging and important are joint interventional effects, where simultaneous interventions on multiple action variables influence a target outcome. Such scenarios are common in fields like epidemiology (Kekić et al., 2023a), e-commerce (Kunz et al., 2023; Schultz et al., 2023), and medicine (Prosperi et al., 2020).

Consider a company trying to optimize their marketing strategy across multiple channels like social media, email campaigns, and display advertising. While the ultimate goal is to understand how these channels work together to drive sales, running experiments for every possible combination of marketing interventions would be prohibitively expensive and time-consuming, as the number of required test conditions grows exponentially with each additional channel. This raises a crucial question: *Can we learn about joint effects from simpler experiments?*

We investigate whether joint interventional effects can be estimated using only observational data and experiments where we intervene on one variable at a time. This is an instance of the *Intervention Generalization Problem* (Bravo-Hermsdorff et al., 2023): predicting treatment effects in previously unseen interventional settings. Causal models encode additional structural relationships between variables that allow us to generalize to settings in which non-causal machine learning approaches assuming independent, identically distributed (i.i.d.) data fail.

However, this problem is not solvable in its most general form. To achieve Intervention Generalization, we need to restrict the causal model class (Saengkyongam & Silva, 2020). The key question becomes: *What model class restrictions enable this generalization while preserving broad applicability?*

In this study, we focus on causal models where each action contributes to the outcome variable in a nonlinear way and is subject to confounding. We show that when these complex individual effects combine additively to produce the outcome, the joint interventional effect is identifiable from observational and single-intervention data. This additivity assumption is well-motivated across several domains—for instance, in pharmacology where many drug combinations exhibit approximately additive effects in the absence of specific interaction mechanisms (Pearson et al., 2023), and in marketing analytics where additive models are widely used to understand how multiple advertising channels influence consumer behavior (Jin et al., 2017; Chan & Perry, 2017).

Our main contributions are:

1. An identifiability result showing that joint interventional effects can be recovered from single-variable interventions and observational data in causal models with a nonlinear additive outcome mechanism

2. A practical estimator that decomposes the causal effect

---

[1]Max Planck Institute for Intelligent Systems, Tübingen, Germany [2]Amazon Research, Tübingen, Germany [3]Tübingen AI Center, Tübingen, Germany [4]ELLIS Institute, Tübingen, Germany. Correspondence to: Armin Kekić <armin.kekic@tue.mpg.de>.

*Proceedings of the 42$^{nd}$ International Conference on Machine Learning*, Vancouver, Canada. PMLR 267, 2025. Copyright 2025 by the author(s).

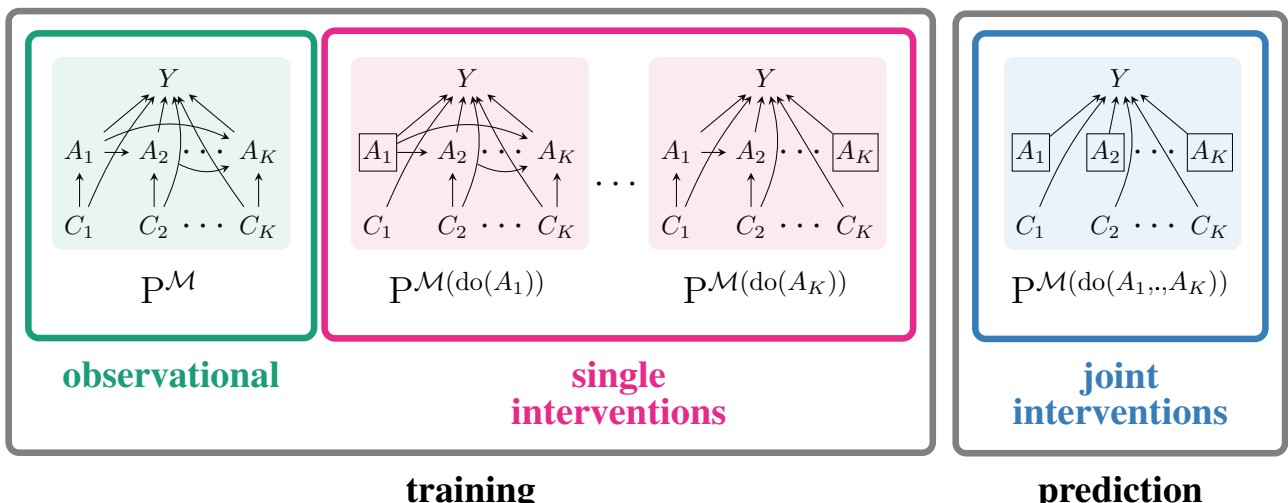

*Figure 1.* **The Intervention Generalization Problem.** The figure shows the different interventional regimes. Our goal is to estimate the joint interventional effect of the action variables $\{A_1, \ldots, A_K\}$ on $Y$, that is, $\mathbb{E}[Y \mid \mathrm{do}(a_1, \ldots, a_K)]$ (right). However, during training we only have access to observational (left) and single-interventional data (middle). There are unobserved confounders $\{C_1, \ldots, C_K\}$ between the actions and the outcome variable $Y$. A box around a variable indicates that it was intervened on.

into confounded and unconfounded contributions for each intervention variable

3. Empirical validation demonstrating that our method achieves performance comparable to models trained directly on joint interventional data

## 2. Problem Statement

### 2.1. Background and Notation

We use boldface for vector-valued or sets of random variables. We denote random variables with capital letters and realizations thereof in lowercase.

**Definition 1** (SCM (Pearl, 2009; Peters et al., 2017))**.** An $N$-dimensional structural causal model is a triplet $\mathcal{M} = (\mathcal{G}, \mathbb{S}, P_{\mathbf{U}})$ consisting of:

- a joint distribution $P_{\mathbf{U}}$ over the jointly independent "exogenous" variables $\mathbf{U} = \{U_1, \ldots, U_N\}$,

- a directed acyclic graph $\mathcal{G}$ with $N$ vertices,

- a set $\mathbb{S} = \{X_j := f_j(\mathbf{Pa}_j, U_j), j = 1, \ldots, N\}$ of structural assignments, where each $f_j$ is a scalar-valued function and $\mathbf{Pa}_j$ are the variables indexed by the set of parents of node $j$ in $\mathcal{G}$,

such that for every $\mathbf{u}$, the system $\{x_j := f_j(\mathbf{pa}_j, u_j)\}$ has a unique solution. The SCM thus entails a joint distribution over the "endogenous" random variables $\mathbf{X} = \{X_1, \ldots, X_N\}$.

**Interventions in SCMs.** Interventions on endogenous variables in SCMs are represented by the $\mathrm{do}(\cdot)$-operator and modify how variables are assigned their values. A *perfect stochastic intervention* $\mathrm{do}(X_i)$ removes all incoming edges to the intervened variable and replaces its structural assignment with a draw from an intervention distribution over $X_i$. That is, we replace the structural assignment $X_k := f_k(\mathbf{Pa}_k, U_k)$ by $X_k := \tilde{U}_k$, where $\tilde{U}_k$ is a univariate distribution independent of the parents $\mathbf{Pa}_k$. We use lowercase letters to denote specific realizations of these interventions, writing $\mathrm{do}(x_i)$ to indicate that $X_i$ was set to the particular value $x_i$. Such interventions transform the original model $\mathcal{M}$ into an interventional model $\mathcal{M}(\mathrm{do}(X_i))$ that induces a modified distribution over the endogenous variables, which we denote with a superscript $P^{\mathcal{M}(\mathrm{do}(X_i))}$. When no superscript is present, the distribution comes from the unintervened (observational) model $\mathcal{M}$. In conditioning sets, we write $\{x_1, \ldots, \mathrm{do}(x_i), \ldots, x_n\}$ to indicate that $X_i$ was intervened on while other variables have observational realizations.

### 2.2. Setting

Let $\mathbf{A} = \{A_1, \ldots, A_K\}$ be a set of *treatment* or *action* variables, $\mathbf{C} = \{C_1, \ldots, C_K\}$ be a set of *unobserved confounders*, and $Y$ be an *outcome* variable. The actions are direct causes of the outcome, and we allow for an arbitrary acyclic causal structure among the actions. For notational simplicity, we assume the actions are in topological order

and write the causal structure as a fully connected DAG.[1]

We can write the structural assignments as follows:

$$Y := f(A_1, \dots, A_K, C_1, \dots, C_K, U) \qquad (1)$$
$$A_k := g_k(A_1, \dots, A_{k-1}, C_1, \dots, C_{k-1}, V_k) \qquad (2)$$
$$C_k := W_k \qquad (3)$$
$$\text{for } k \in \{1, \dots, K\}$$

where $\{U, V_1, \dots, V_K, W_1, \dots, W_K\}$ are mutually independent exogenous noise variables.

We are given a dataset of observational i.i.d. samples

$$D_{\text{obs}} \sim P^{\mathcal{M}}_{(Y, \mathbf{A})} \qquad (4)$$

and $K$ datasets of i.i.d. samples for single-variable interventions on each action variable

$$\{D^k_{\text{int}} \sim P^{\mathcal{M}(\text{do}(A_k))}_{(Y, \mathbf{A})}\}_{k=1}^K. \qquad (5)$$

Here, each interventional dataset $D^k_{\text{int}}$ consists of samples generated under perfect stochastic interventions on $A_k$.

Our objective is to estimate the *joint interventional effect*

$$\mathbb{E}\left[Y \mid \text{do}(a_1, \dots, a_K)\right]. \qquad (6)$$

**Identifiability.** For a given class of causal models, we say that a causal effect is identifiable from a set of probability distributions if it can be uniquely determined from these distributions. In our setting, we say that the joint interventional effect (6) is identifiable from observational and single-intervention distributions if it can be uniquely determined from $P^{\mathcal{M}}_{(Y, \mathbf{A})}$ and $\{P^{\mathcal{M}(\text{do}(A_k))}_{(Y, \mathbf{A})}\}_{k=1}^K$. Note that this is equivalent to saying that the joint interventional effect (6) is identifiable from infinite samples from $P^{\mathcal{M}}_{(Y, \mathbf{A})}$ and $\{P^{\mathcal{M}(\text{do}(A_k))}_{(Y, \mathbf{A})}\}_{k=1}^K$, *i.e.* when the datasets $D_{\text{obs}}$ and $\{D^k_{\text{int}}\}_{k=1}^K$ are infinitely large.

Conversely, the joint interventional effect is not identifiable from single-interventional and observational distributions within a given model class if there exist two distinct causal models that entail the same single-interventional and observational distributions but different joint-interventional distributions.

## 3. Related Work

Understanding causal effects from data with limited experimental control is a fundamental challenge in causal inference. There is a rich literature that studies such problems

in the *nonparametric setting*, where only knowledge about the causal structure and probability distributions in different interventional settings are given. Such methods are agnostic to the functional form of the causal mechanisms and the distribution of the exogenous noise variables. The foundational work in this area focuses on identifying interventional effects from purely observational data (Tian & Pearl, 2002; Shpitser & Pearl, 2006). The multi-treatment setting, where effects of multiple simultaneous treatments must be estimated from purely observational data, presents additional challenges (Miao et al., 2023; Zheng et al., 2025). This has been generalized through *g-identification* theory (Lee et al., 2019), which determines whether target interventional effects can be recovered from a given combination of observational and experimental data. However, as we show in Section 4, nonparametric approaches are insufficient for our setting. This motivates our focus on parametric assumptions to identify the joint interventional effect (6).

Bravo-Hermsdorff et al. (2023) present a *factor graph* approach for the Intervention Generalization Problem, investigating identifiability of joint interventional effects given specific factorizations of observational and interventional probability distributions. Similarly, Jung et al. (2023) present graphical conditions for nonparametric identification of joint interventional effects—which they term *Multiple Treatment Interactions*—and employ double machine learning techniques for estimation from marginal interventional data. Related approaches study causal effects in novel action-context pairs when actions and contexts are categorical variables (Ribot et al., 2024) or and treatment effect estimation for unobserved subgroups in causal mixture models (Mazaheri et al., 2024).

The most closely related prior work is that of Saengkyongam & Silva (2020), who employ parametric assumptions on the exogenous noise variables. They show that generalization from single-intervention and observational data to the joint interventional effect is possible for continuous variables when the exogenous noise is Gaussian and additive. In contrast, we consider parametric assumptions on the causal mechanism (1) that relates the action variables to the outcome variable, assuming that it is additive. While this restricts the confounding between the actions, this allows us to treat any distributions for the exogenous noise as well as both discrete and continuous variables—see Appendix A.1.

*Additive models* have a long history in statistics and machine learning. They were first introduced for regression (Friedman & Stuetzle, 1981) and later extended to *Generalized Additive Models (GAMs)* (Breiman & Friedman, 1985; Hastie & Tibshirani, 1990), where the additive contributions from each input variable are transformed through a nonlinear link function. More recently, *Neural Additive Models* have combined the interpretability benefits of GAMs with

---

[1]This means that each action variable $A_k$ can potentially depend on all preceding actions $\{A_1, \dots, A_{k-1}\}$ and confounders $\{C_1, \dots, C_{k-1}\}$.

the flexibility of neural networks (Agarwal et al., 2021). The concept of additivity has also enabled novel forms of generalization—Lachapelle et al. (2023) showed that non-linear additive ground-truth decoders allow generalization to unseen combinations of latent factors, which they termed *Cartesian-Product Extrapolation*. While additive noise assumptions are common in causal inference (Hoyer et al., 2008), this is, to the best of our knowledge, the first work that investigates additivity of causal mechanisms with respect to the parent variables.

The complementary problem of generalizing from joint interventions to single intervention effects was studied by Jeunen et al. (2022) and Elahi et al. (2024), and generalization to unseen interventions without identifiability was explored for stationary diffusion models (Lorch et al., 2024).

Intervention Generalization is akin to other types of generalization, which is aided by the structure encoded in causal models such as the *Causal Marginal Problem* (Gresele et al., 2022; Garrido Mejia et al., 2022; 2024). There, instead of generalizing to a new interventional regime, the goal is to learn about the joint behavior and causal structure of variables that have only been observed in subsets, but never jointly. Another example is *Out-of-Variable Generalization* (Guo et al., 2024), where some variables have never been observed in training.

A key aspect of our method is the ability of causal models to combine information from different data sets. This aligns with *Causal Representation Learning*, where many approaches use datasets or observation pairs that differ by interventions in the latent variables (Liang et al., 2023; Brehmer et al., 2022; Zhang et al., 2023; von Kügelgen et al., 2023; Buchholz et al., 2023), samples from scientific simulations (Kekić et al., 2023a;b), and multiple views and modalities (Yao et al., 2024).

## 4. General Non-Identifiability

In general, the setting described in Section 2.2 is not identifiable. We can show that two distinct SCMs can induce identical observational distributions and single-variable interventional distributions, but exhibit different behaviors under joint interventions.

*Example* 1. Consider the following two SCMs over binary variables:

$\mathcal{M} :$

$$Y := A_1 \wedge A_2 \wedge C \wedge U$$
$$A_2 := A_1 \wedge C \wedge V_2$$
$$A_1 := C$$
$$C := W$$

$\widetilde{\mathcal{M}} :$

$$Y := A_2 \wedge C \wedge U$$
$$A_2 := A_1 \wedge C \wedge V_2$$
$$A_1 := C$$
$$C := W$$

where $U, V_2, W \sim \text{Bernoulli}(p)$, with $0 < p < 1$. These two models induce the same observational distribution over the observed variables; that is, $\mathrm{P}^{\mathcal{M}}(Y, A_1, A_2) = \mathrm{P}^{\widetilde{\mathcal{M}}}(Y, A_1, A_2)$. They also lead to the same single-variable interventional distributions; $\mathrm{P}^{\mathcal{M}(\text{do}(A_1))}(Y, A_1, A_2) = \mathrm{P}^{\widetilde{\mathcal{M}}(\text{do}(A_1))}(Y, A_1, A_2)$ and $\mathrm{P}^{\mathcal{M}(\text{do}(A_2))}(Y, A_1, A_2) = \mathrm{P}^{\widetilde{\mathcal{M}}(\text{do}(A_2))}(Y, A_1, A_2).$[2] However, they induce different distributions when $A_1$ and $A_2$ are jointly intervened:

$$\mathrm{P}^{\mathcal{M}(\text{do}(A_1=0, A_2=1))}(Y=1) = 0$$
$$\neq p^2 = \mathrm{P}^{\widetilde{\mathcal{M}}(\text{do}(A_1=0, A_2=1))}(Y=1) . \qquad (7)$$

Hence, two estimators trained on observational and single-interventional data from $\mathcal{M}$ and $\widetilde{\mathcal{M}}$ would arrive at identical predictions for the joint interventional effect (6), despite the true effects being different in these two models.

## 5. Assumptions

In the previous section, we demonstrated that joint interventional effects cannot be identified from single-variable interventions in general. However, by imposing certain restrictions on the causal model class, we can achieve identifiability. In this section, we introduce two key assumptions that together enable the identification of joint interventional effects from single-variable interventions.

**Assumption 1** (Intervention Support). The distributions of the action variables have identical support across all interventional regimes. That is,[3]

$$\text{supp}_{\mathrm{P}^{\mathcal{M}}_{(\mathbf{A})}}(\mathbf{A}) = \text{supp}_{\mathrm{P}^{\mathcal{M}(\text{do}(A_1,..,A_K))}_{(\mathbf{A})}}(\mathbf{A})$$
$$= \text{supp}_{\mathrm{P}^{\mathcal{M}(\text{do}(A_k))}_{(\mathbf{A})}}(\mathbf{A}) \quad \text{for any } k \in \{1, ..., K\}. \qquad (8)$$

**Assumption 2** (Additive Outcome Mechanism). There is pair-wise confounding between the actions and the outcome. The outcome is generated by an additive combination of separate nonlinear functions for each action and its associated confounder. The structural assignments can be written as:

$$Y := \sum_{k=1}^{K} f_k(A_k, C_k) + U \qquad (9)$$
$$A_k := g_k(A_1, ..., A_{k-1}, C_k, V_k) \qquad (10)$$
$$C_k := W_k \qquad (11)$$
$$\text{for } k \in \{1, ..., K\}$$

where $\{U, V_1, ..., V_K, W_1, ..., W_K\}$ are mutually independent exogenous noise variables.

---

[2]These probability distributions are shown in Tables 1 to 3 in Appendix D.

[3]We denote the support of random variable $\mathbf{A}$ under the probability distribution $\mathrm{P}^{\mathcal{M}}_{(\mathbf{A})}$ as $\text{supp}_{\mathrm{P}^{\mathcal{M}}_{(\mathbf{A})}}(\mathbf{A})$. Where $\mathcal{M}$ is the corresponding causal model; in this case, the SCM for the observational regime.

Assumption 1 is needed for technical reasons in the identifiability proof. In order to identify the joint effect (6), we need to transfer information from the other interventional regimes (4) and (5) to the joint case (6). Assumption 1 ensures that the actions cover the same range for the inputs of the outcome mechanism (1). Such assumptions on the support of variables are common in multi-dataset settings in Causal Representation Learning (Varici et al., 2024) and can be seen as a version of the positivity assumption in causal effect estimation (Hernan & Robins, 2010, Chapter 3.3). In practice, identical intervention support can be ensured through choosing the appropriate experimental settings. When this is not possible, support mismatches can be mitigated by exploiting domain knowledge about the function classes in the structural assignments (Kekić et al., 2023a); in our case $\{f_k\}_{k=1}^K$.

Assumption 2 restricts the causal model class regarding how the actions influence the outcome. While the effects of each action $A_k$ on the outcome $Y$ can be nonlinear and complex, we limit how the actions and confounders interact in the outcome mechanism (1). We assume that both the contributions of each action-confounder pair $(A_k, C_k)$ and the exogenous noise $U$ are additive. Note that for the other structural assignments we do not introduce such constraints for the exogenous noise or functional form. The resulting causal structure is illustrated in Figure 1 (left).

## 6. Theoretical Results

### 6.1. Identifiability of Joint Interventional Effect

In this section, we show that in the causal model class with a nonlinear additive outcome mechanism, outlined in Section 5, we can achieve Intervention Generalization.

**Theorem 1** (Identifiability). *Under the assumptions in Section 5, the joint interventional effect (6) is identifiable from single-variable interventions and observational data in the infinite data regime.*

**Proof Sketch** We first note that we can decompose the joint interventional effect (6), which we want to estimate, as well as the conditional expectations, for which we have data, as

$$\mathbb{E}[Y \mid \mathrm{do}(a_1, \dots, a_K)]$$
$$= \sum_k \mathbb{E}_{C_k \sim \mathrm{p}(C_k)} [f_k(a_k, C_k)] \qquad (12)$$

$$\mathbb{E}[Y \mid a_1, \dots, \mathrm{do}(a_j), \dots, a_K]$$
$$= \mathbb{E}_{C_j \sim \mathrm{p}(C_j)}[f_j(a_j, C_j)]$$
$$\quad + \sum_{k \neq j} \mathbb{E}_{C_k \sim \mathrm{p}(C_k|a_1, ., a_K)}[f_k(a_k, C_k)]$$
$$\text{for } j \in \{1, \dots, K\} \qquad (13)$$

$$\mathbb{E}[Y \mid a_1, \dots, a_K]$$
$$= \sum_k \mathbb{E}_{C_k \sim \mathrm{p}(C_k|a_1, ., a_K)}[f_k(a_k, C_k)]. \qquad (14)$$

These decompositions correspond to the terms $f_k$ in the additive outcome mechanism (9). In each term, the expectation over the confounder $C_k$ is taken with respect to a measure that depends on whether the corresponding action variable $A_k$ was intervened on. When $A_k$ is not intervened on, the confounding can introduce an additional dependence on the other action variables, entangling their influences.

However, these decompositions still allow us to learn a representation that enables us to generalize from the observational and single-interventional setting to the joint-interventional effect. We define $K$ estimator functions

$$\hat{f}_k(a_1, \dots, a_K, R_k), \qquad k \in \{1, \dots, K\}, \qquad (15)$$

where $R_k \in \{0, 1\}$ indicates an intervention on $A_k$. Each $R_k$ can be thought of as selecting one of two functions

$$\hat{f}_k(a_1, \dots, a_K, R_k) = \begin{cases} \hat{f}_k^{\mathrm{obs}}(a_1, \dots, a_K), \text{ if } R_k{=}0 \\ \hat{f}_k^{\mathrm{int}}(a_1, \dots, a_K), \text{ if } R_k{=}1 \end{cases}$$
$$(16)$$

where $\hat{f}_k^{\mathrm{int}}$ represent the terms in the decompositions (12)–(14) where the corresponding action $A_k$ is intervened on, and $\hat{f}_k^{\mathrm{obs}}$ are the factors with $A_k$ observational. We then define an overall estimator

$$\hat{f}(a_1, \dots, a_K, R_1, \dots, R_K) = \sum_{k=1}^K \hat{f}_k(a_1, \dots, a_K, R_k)$$
$$(17)$$

to represent all regimes, depending on the setting of the indicator variables $R_1, \dots, R_K$:

- When $R_1{=}1, \dots, R_K{=}1$, the function $\hat{f}$ is an estimator for the joint interventional regime.

- $R_1{=}0, \dots, R_j{=}0, \dots, R_K{=}1$ corresponds to the single-intervention setting of $\mathcal{M}(\mathrm{do}(a_j))$.

- $R_1{=}0, \dots, R_K{=}0$ is the observational setting.

In the outcome mechanism (9), each term $f_k$ depends only on one action $A_k$.[4] In contrast, each model factor $\hat{f}_k$ has to take all actions into account due to the entanglement introduced through confounding.

Now if we fit the estimator $\hat{f}$ in the observational and the single-interventional regimes, that is,

$$\hat{f}(a_1, \dots, a_K, R_1{=}0, \dots, R_K{=}0) = \mathbb{E}[Y \mid a_1, \dots, a_K]$$
$$(18)$$

---

[4] $f_k$ also depends on the confounder $C_k$.

$$\hat{f}(a_1, \ldots, a_K, R_1{=}0, \ldots, R_j{=}1, \ldots, R_K{=}0)$$
$$= \mathbb{E}[Y \mid a_1, \ldots, \mathrm{do}(a_j), \ldots, a_K], \quad \text{for } j \in \{1, \ldots, K\},$$
$$\tag{19}$$

we can show that the estimator also identifies the joint interventional effect:

$$\hat{f}(a_1, \ldots, a_K, R_1{=}1, \ldots, R_K{=}1) = \mathbb{E}[Y \mid \mathrm{do}(a_1, \ldots, a_K)]. \tag{20}$$

The full proof is shown in Appendix A. ∎

Note that, while our approach assumes that the action variables are direct causes of the outcome and that there is no confounding between the actions, we do not assume a particular causal structure between the actions. The approach we present here is agnostic to causal relationships among the actions and does not use this information to infer the joint interventional effect (6).

### 6.2. Mixed Interventional Effects

Moreover, we can extend our results to any combination of intervened and observational actions:

**Proposition 1** (Identifiability of Mixed Interventional Effects). *Let* $\mathbf{A}_{\mathrm{int}} \cup \mathbf{A}_{\mathrm{obs}} = \{A_1, \ldots, A_K\}$ *be a partition of the action variables into intervened and observational actions. Under the assumptions in Section 5, the effect*

$$\mathbb{E}[Y \mid \mathrm{do}(\mathbf{a}_{\mathrm{int}}), \mathbf{a}_{\mathrm{obs}}] \tag{21}$$

*is identifiable from single-variable interventions and observational data in the infinite data regime. The proof is given in Appendix B.*

### 6.3. Additivity with Respect to a Partition

Theorem 1 implies that for an additive outcome mechanism (1), the number of interventional datasets required for identification of the joint effect (6) grows only linearly with the number of actions. We can show that even when (1) is only additive with respect to the effect of subsets of actions, identification is possible as long as we have joint interventional data on each subset.

**Definition 2** (Additivity w.r.t. a Partition). Let $\mathfrak{B}$ be a partition of the index set $\{1, \ldots, K\}$. For an SCM of the form discussed in Section 2.2, we call the outcome mechanism (1) *additive with respect to* $\mathfrak{B}$, if we can write the structural assignments as:

$$Y := \sum_{B \in \mathfrak{B}} f_B(\mathbf{A}_B, \mathbf{C}_B) + U \tag{22}$$

$$A_k := g_k(A_1, \ldots, A_{k-1}, \mathbf{C}_{B(k)}, V_k) \tag{23}$$

$$C_k := W_k \tag{24}$$

$$\text{for } k \in \{1, \ldots, K\},$$

where $B(k) \in \mathfrak{B}$ is the partition that contains index $k$.

**Corollary 1.** *Let* $\mathfrak{B}$ *be a partition of the index set* $\{1, \ldots, K\}$ *such that the ground-truth SCM* $\mathcal{M}$ *has an outcome mechanism which is additive with respect to* $\mathfrak{B}$. *Then the joint interventional effect* (6) *can be identified from observational data*

$$\mathrm{D}_{\mathrm{obs}} \sim \mathrm{P}_{(Y, \mathbf{A})}^{\mathcal{M}} \tag{25}$$

*and* $|\mathfrak{B}|$ *interventional datasets*

$$\{\mathrm{D}_{\mathrm{int}}^B \sim \mathrm{P}_{(Y, \mathbf{A})}^{\mathcal{M}(\mathrm{do}(\mathbf{A}_B))}\}_{B \in \mathfrak{B}}. \tag{26}$$

*The proof is given in Appendix C.*

Hence, we can trade off assumptions of additivity on the outcome mechanism for joint experimentation on actions in two complementary ways. First, if it is unclear whether the outcome mechanism decomposes per variable within some subsets of actions, we can choose not to make the additivity assumption on these subsets and instead collect joint interventional data on them. Second, our framework allows for shared confounding among action variables within the same partition subset, provided we collect joint interventional data on that subset. In other words, while our base assumption requires pair-wise confounding (each confounder $C_k$ affects only action $A_k$ and the outcome), Corollary 1 permits more complex confounding structures within partition subsets—for instance, a single confounder could affect multiple actions within the same subset—as long as we have joint interventional data for that subset. This flexibility makes our method applicable to a broader range of real-world scenarios where strict additivity or simple confounding structures may not hold across all variables, while still maintaining identifiability guarantees through strategic experimental design.

## 7. Experiments

### 7.1. Setting

**Synthetic Data-Generating Process.** We sample a structural causal model with five actions and confounders and causal relationships as shown in Figure 3. The structural assignments are second order polynomials with randomly sampled coefficients. The exogenous noises are Gaussian, uniform or logistic. The corresponding parameters are drawn at random before each experiment run. The dependencies between actions are probabilistic, with each potential edge having a probability 0.3 of being active. We sample 100 SCMs, where for each run we sample $N_{\mathrm{obs}}$, $N_{\mathrm{sint}}$ and $N_{\mathrm{jint}}$ data points for the observational, the single-interventional- and joint-interventional datasets. We split each dataset into 80% training- and 20% test data. For evaluation, all models are tested on the joint interventional test dataset. Further details about the experimental setup are given in Appendix E.

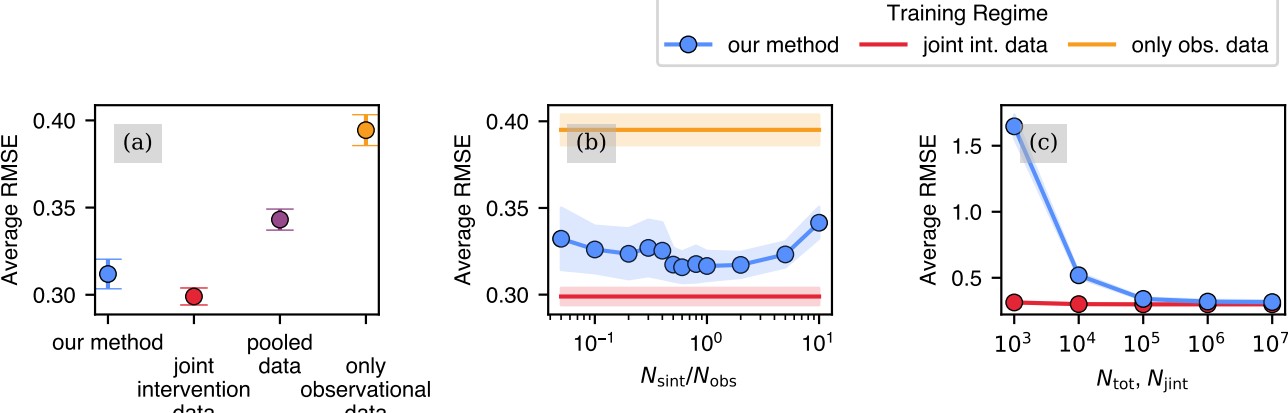

*Figure 2.* **Experiments on Synthetic Data. (a)** Average root mean squared error (RMSE) for predicting the joint interventional effect $\mathbb{E}[Y \mid \mathrm{do}(a_1, \dots, a_5)]$, averaged over 100 experiment runs. Each run uses a randomly generated ground truth SCM. We compare three approaches: (i) Our Intervention Generalization method, training the estimator (17) on observational and single-intervention data (Section 6). (ii) An estimator trained directly on joint interventional data (topline). (iii) A naive estimator trained on pooled dataset of all observational and single-interventional data. (iv) An estimator trained solely on observational data. The error bars show the standard error of the mean. **(b)** Prediction error of the joint interventional effect (6) under varying ratios of observational and single-interventional data for a fixed number of total data points. **(c)** Prediction error of (6) with increasing total number of data points. The full experimental details are given in Appendix E.

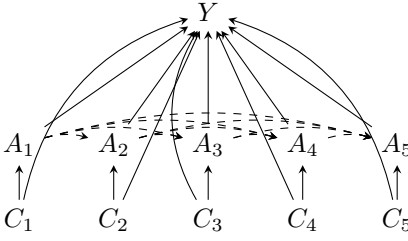

*Figure 3.* **Causal Graph in Synthetic Experiments.** Dashed edges between actions represent probabilistic dependencies that may or may not exist in each sampled SCM.

**Models and Benchmarks.** We train third order polynomial estimator functions (17) as outlined in Section 6. We compare that model to three baselines.

(i) A model that is directly trained on joint interventional data. That is, we directly fit $\mathbb{E}[Y \mid \mathrm{do}(a_1, \dots, a_5)]$. This comparison represents the minimal error that our method could achieve.

(ii) A naive estimator directly trained on the pooled single-interventional and observational data.

(iii) A model that only considers the observational data. That is, we use a model fit to $\mathbb{E}[Y \mid a_1, \dots, a_5]$ to predict $\mathbb{E}[Y \mid \mathrm{do}(a_1, \dots, a_5)]$. This is a typical approximation made in the absence of interventional data in

real-world applications (Kunz et al., 2023).[5]

The full experimental details and additional results are given in Appendix E.

### 7.2. Results

The mean root mean squared error (RMSE) over all sampled SCMs in each of the four estimation methods is shown in Figure 2(a). We observe that our method achieves an error that is close to the minimally achievable error of the topline model that is directly trained on joint interventional data. Both our approach and the topline benchmark significantly outperform the naive observational-only model and the estimator trained on pooled data. The results empirically validate the effectiveness of our Intervention Generalization technique in leveraging single-intervention data to predict joint-interventional effects.

A practically relevant question is: *How much single interventional data is necessary to obtain a good estimate of the joint effect* (6)? Typically, interventional data is much harder to obtain than observations of the unintervened system. We empirically test the behavior of the Intervention Generalization approach with varying shares of single-interventional and observational data, shown in Figure 2(b). That is, the total number of data points $N_{\mathrm{tot}} = N_{\mathrm{obs}} + K N_{\mathrm{sint}}$ is fixed, and the ratio $N_{\mathrm{sint}}/N_{\mathrm{obs}}$ is changed. The precise value of

---

[5]The additional error incurred through making this simplifying assumption quantifies the *Causal Risk* (Vankadara et al., 2022).

the optimal ratio depends on the overall number of data points and the number of actions in the SCM. We observe two trends: (i) The optimal ratio tends to be between 0.1 and 0.9. Thus, having more observational data is favorable. This aligns well with real-world scenarios where observational data is typically more abundant and easier to collect. (ii) The error curve flattens as $N_{tot}$ increases. Additional experimental runs for more settings are shown in Figure B in Appendix E.1.

The theoretical results in Section 6 establish that the joint interventional effect (6) can be identified from single-interventional and observational data. However, from those results we cannot infer the sample efficiency of the estimation procedure in the proof of Theorem 1. Figure 2(c) shows the error as a function of the number of total data points $N_{tot}$. We compare this against an estimator that is directly trained on a dataset of joint interventions of size $N_{jint}$. We find that the Intervention Generalization approach takes over an order of magnitude more data to reach a similar accuracy as the model directly trained on joint interventions. Hence, the benefit of requiring only single interventions comes at the cost of a decreased sample efficiency and has to be weighed against the difficulty of obtaining joint interventional data. Figure A in Appendix E.1 shows the convergence for SCMs with varying numbers of actions $K$.

## 8. Discussion and Outlook

We have shown that by constraining the outcome mechanism (1) to an additive model class, we can successfully identify joint interventional effects from single-interventional and observational data. Our constructive identifiability proof provides a practical estimator for the joint interventional effect (6). The estimator function decomposes into terms for the confounded and unconfounded contribution of each action to the outcome.

While additivity offers mathematical tractability, careful consideration is needed before applying this assumption. In ecological systems, for instance, Brook et al. (2008) showed that environmental threats often interact synergistically rather than additively, where the combined impact of multiple stressors exceeded the sum of their individual effects, leading to systematic underestimation of extinction risks when additive models were used.

Our identifiability result based on additivity constraints points to a broader theoretical question: *What is the minimal set of assumptions required for identifiability, and can we find more general causal model classes that still permit identification while relaxing our current constraints?* Identifying such classes would expand the practical applicability of our approach while maintaining its theoretical guarantees.

One avenue for generalizing the results of this work is the function class of the outcome mechanism (1). At the most general end of the spectrum is the additive structure used in the Kolmogorov–Arnold representation theorem (Kolmogorov, 1957; 1956), which can represent any continuous function from a compact set in $\mathbb{R}^N$ to $\mathbb{R}$. However, with such broad representational power, we are unlikely to maintain identifiability. Moving toward more restricted classes, Generalized Additive Functions (Hastie & Tibshirani, 1990) and Post-Nonlinear Models (Zhang & Hyvärinen, 2009) represent promising intermediate points that could balance expressiveness with structural constraints. The latter have already demonstrated utility in causal structure learning. Such intermediate function classes could extend our approach to scenarios where strict additivity may not hold.

The precise confounding structure can be difficult to assess and often there are additional covariates to account for. Another area for investigation is thus the adaptation of our estimation technique to more complex scenarios. These include settings with additional non-intervened covariates or under more general confounding structures between action variables.

While identifiability guarantees that the joint effect can be estimated without joint interventional data, we have observed that this estimate can be inefficient in terms of sample complexity compared to directly training on joint interventions. Therefore, another direction of future work could be to investigate finite sample guarantees of Intervention Generalization.

## Acknowledgements

We would like to thank Simon Buchholz, Amartya Sanyal and Frederik Träuble for their valuable contributions through interesting and helpful discussions about our approach.

## Software and Data

The code used for the experiments is available at github.com/akekic/intervention-generalization.

## Impact Statement

This paper presents work whose goal is to advance the field of Machine Learning. There are many potential societal consequences of our work, none which we feel must be specifically highlighted here.

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

# A. Proof of Theorem 1

## A.1. A Note on Variable Types

For notational simplicity, we present all proofs using continuous variables and the Lebesgue measure. However, our results hold more generally for discrete variables or a mix of discrete and continuous variables. The proofs can be adapted by replacing the Lebesgue measure with an appropriate measure for each variable type—the counting measure for discrete variables and the Lebesgue measure for continuous variables. All integrals would then be interpreted with respect to these measures, effectively becoming sums for discrete variables.

The remaining assumptions and proof steps carry through with these modifications, as the key properties we rely on—such as the law of total probability and the independence relations in the causal graph—hold under general probability measures.

## A.2. Lemmas

Before we prove the main proposition, we introduce two useful lemmas.

**Lemma 1.** *Let $\mathcal{M}$ be an SCM as defined above. Then,*

$$\mathrm{p}^{\mathcal{M}(\mathrm{do}(a_j))}(c_k \mid a_1, \dots, \mathrm{do}(a_j), \dots, a_k) = \mathrm{p}(c_k \mid a_1, \dots, a_j, \dots, a_k), \tag{27}$$

*for all $k \in \{2, \dots, K\}$, and $j \in \{1, \dots, k-1\}$.*

**Proof** (Lemma 1) Using Bayes' rule on the left-hand side of Equation (27) we have,

$$\mathrm{p}^{\mathcal{M}(\mathrm{do}(a_j))}(c_k \mid a_1, \dots, \mathrm{do}(a_j), \dots, a_k) \tag{28}$$

$$= \underbrace{\mathrm{p}^{\mathcal{M}(\mathrm{do}(a_j))}(a_k \mid a_1, \dots, \mathrm{do}(a_j), \dots, a_{k-1}, c_k)}_{\text{causal mechanism}} \dfrac{\overbrace{\mathrm{p}^{\mathcal{M}(\mathrm{do}(a_j))}(c_k \mid a_1, \dots, \mathrm{do}(a_j), \dots, a_{k-1})}^{\text{no open path between } C_k \text{ and } A_1, ., \mathrm{do}(A_j), ., A_{k-1}}}{\mathrm{p}^{\mathcal{M}(\mathrm{do}(a_j))}(a_k \mid a_1, \dots, \mathrm{do}(a_j), \dots, a_{k-1})} \tag{29}$$

$$= \mathrm{p}(a_k \mid a_1, \dots, a_j, \dots, a_{k-1}, c_k) \dfrac{\overbrace{\mathrm{p}^{\mathcal{M}(\mathrm{do}(a_j))}(c_k)}^{\text{root node}}}{\mathrm{p}^{\mathcal{M}(\mathrm{do}(a_j))}(a_k \mid a_1, \dots, \mathrm{do}(a_j), \dots, a_{k-1})} \tag{30}$$

$$= \mathrm{p}(a_k \mid a_1, \dots, a_j, \dots, a_{k-1}, c_k) \dfrac{\mathrm{p}(c_k)}{\mathrm{p}^{\mathcal{M}(\mathrm{do}(a_j))}(a_k \mid a_1, \dots, \mathrm{do}(a_j), \dots, a_{k-1})}. \tag{31}$$

We now focus on the denominator,

$$\mathrm{p}^{\mathcal{M}(\mathrm{do}(a_j))}(a_k \mid a_1, \dots, \mathrm{do}(a_j), \dots, a_{k-1}) \tag{32}$$

$$= \int \mathrm{p}^{\mathcal{M}(\mathrm{do}(a_j))}(a_k, c_k \mid a_1, \dots, \mathrm{do}(a_j), \dots, a_{k-1}) \, \mathrm{d}c_k \tag{33}$$

$$= \int \mathrm{p}^{\mathcal{M}(\mathrm{do}(a_j))}(a_k \mid a_1, \dots, \mathrm{do}(a_j), \dots, a_{k-1}, c_k) \, \mathrm{p}^{\mathcal{M}(\mathrm{do}(a_j))}(c_k \mid a_1, \dots, \mathrm{do}(a_j), \dots, a_{k-1}) \, \mathrm{d}c_k \tag{34}$$

$$= \int \mathrm{p}(a_k \mid a_1, \dots, a_j, \dots, a_{k-1}) \, \mathrm{p}^{\mathcal{M}(\mathrm{do}(a_j))}(c_k) \, \mathrm{d}c_k \tag{35}$$

$$= \int \mathrm{p}(a_k \mid a_1, \dots, a_j, \dots, a_{k-1}) \, \mathrm{p}(c_k) \, \mathrm{d}c_k \tag{36}$$

$$= \int \mathrm{p}(a_k \mid a_1, \dots, a_j, \dots, a_{k-1}) \, \mathrm{p}(c_k \mid a_1, \dots, a_j, \dots, a_{k-1}) \, \mathrm{d}c_k \tag{37}$$

$$= \mathrm{p}(a_k \mid a_1, \dots, a_j, \dots, a_{k-1}). \tag{38}$$

Using Equation (38) on Equation (31), and using Bayes' rule we obtain,

$$\mathrm{p}(a_k \mid a_1, \dots, a_j, \dots, a_{k-1}, c_k) \dfrac{\mathrm{p}(c_k \mid a_1, \dots, a_j, \dots, a_{k-1})}{\mathrm{p}(a_k \mid a_1, \dots, a_j, \dots, a_{k-1})} = \mathrm{p}(c_k \mid a_1, \dots, a_j, \dots, a_k), \tag{39}$$

as required. ∎

**Lemma 2.** *The following identities hold:*

*a)* $\mathrm{p}^{\mathcal{M}(\mathrm{do}(a_1,..,a_K))}(c_1,...,c_K \mid \mathrm{do}(a_1,...,a_K)) = \prod_k \mathrm{p}(c_k)$.

*b)* $\mathrm{p}^{\mathcal{M}(\mathrm{do}(a_j))}(c_1,...,c_k \mid a_1,...,\mathrm{do}(a_j),...,a_K) = \mathrm{p}(c_j) \prod_{k \neq j} \mathrm{p}(c_k \mid a_1,...,a_k)$

*c)* $\mathrm{p}(c_1,...,c_k \mid a_1,...,a_k) = \prod_k \mathrm{p}(c_k \mid a_1,...,a_k)$

**Proof** (Lemma 2)

a) Given the causal graph, and since all action variables $A_k$ are intervened on, the only way in which we could introduce dependencies between the confounders is through conditioning on the collider $Y$. Hence, $C_i \perp\!\!\!\perp C_j \mid \mathrm{do}(A_1,...,A_k)$, for all $i,j \in \{1,...,K\}$. Thus, $\mathrm{p}(c_1,...,c_K \mid \mathrm{do}(a_1,...,a_k)) = \prod_k \mathrm{p}(c_k \mid \mathrm{do}(a_1,...,a_k))$. Furthermore, since intervention cuts the dependence to the action variables, and $Y$ is not conditioned on, we have $C_K \perp\!\!\!\perp \mathrm{do}(A_1,...,A_K)$ for all $K$, giving us the first identity.

b) We have $C_i \perp\!\!\!\perp C_k \mid A_1,...,\mathrm{do}(A_j),...,A_k$ for all $i,k \in \{1,...,K\}$ since the conditioning set blocks all paths between the confounders. Either $A_k$ block the outgoing path from $C_k$ for unintervened actions, or there is no outgoing edge (other than to $Y$) for $C_j$.

Additionally, we have $C_j \perp\!\!\!\perp A_1,...,\mathrm{do}(A_j),...,A_K$. Hence,

$$\mathrm{p}^{\mathcal{M}(\mathrm{do}(a_j))}(c_1,...,c_K \mid a_1,...,\mathrm{do}(a_j),...,a_K) \tag{40}$$

$$= \overbrace{\mathrm{p}^{\mathcal{M}(\mathrm{do}(a_j))}(c_j)}^{\text{root node}} \prod_{k \neq j} \mathrm{p}^{\mathcal{M}(\mathrm{do}(a_j))}(c_k \mid a_1,...,\mathrm{do}(a_j),...,a_K) \tag{41}$$

$$= \mathrm{p}(c_j) \prod_{k \neq j} \mathrm{p}^{\mathcal{M}(\mathrm{do}(a_j))}(c_k \mid a_1,...,\mathrm{do}(a_j),...,a_K) \tag{42}$$

$$= \mathrm{p}(c_j) \prod_{k \neq j} \mathrm{p}(c_k \mid a_1,...,a_K), \tag{43}$$

where in the last line we use Lemma 1.

c) Follows from an argument analogous to b).

∎

## A.3. Proof of main result

**Proof** (Theorem 1) First note that

a)

$$\mathbb{E}[Y \mid \mathrm{do}(a_1,...,a_K)] = \int y\, \mathrm{p}^{\mathcal{M}(\mathrm{do}(a_1,..,a_K))}(y \mid \mathrm{do}(a_1,...,a_K))\, \mathrm{d}y \tag{44}$$

$$= \int \cdots \int y\, \mathrm{p}^{\mathcal{M}(\mathrm{do}(a_1,..,a_K))}(y \mid \mathrm{do}(a_1,...,a_K),c_1,...,c_K)\, \mathrm{d}y$$
$$\times \mathrm{p}^{\mathcal{M}(\mathrm{do}(a_1,..,a_K))}(c_1,...c_K \mid \mathrm{do}(a_1,...,a_K))\, \mathrm{d}c_1 \,...\, \mathrm{d}c_K \tag{45}$$

$$= \int \cdots \int \mathbb{E}[Y \mid \mathrm{do}(a_1,...,a_K),c_1,...,c_K]$$
$$\times \mathrm{p}^{\mathcal{M}(\mathrm{do}(a_1,..,a_K))}(c_1,...c_K \mid \mathrm{do}(a_1,...,a_K))\, \mathrm{d}c_1 \,...\, \mathrm{d}c_K \tag{46}$$

$$= \int \cdots \int \left( \sum_k f_k(a_k,c_k) \right) \mathrm{p}^{\mathcal{M}(\mathrm{do}(a_1,..,a_K))}(c_1,...c_K \mid \mathrm{do}(a_1,...,a_K))\, \mathrm{d}c_1 \,...\, \mathrm{d}c_K \tag{47}$$

$$\overset{\text{Lemma 2a)}}{=} \int \ldots \int \left( \sum_k f_k(a_k, c_k) \right) \prod_k \mathrm{p}(c_k) \, \mathrm{d}c_1 \ldots \mathrm{d}c_K \tag{48}$$

$$= \sum_k \int f_k(a_k, c_k) \, \mathrm{p}(c_k) \, \mathrm{d}c_k \tag{49}$$

$$= \sum_k \mathbb{E}_{C_k \sim \mathrm{p}(C_k)} \left[ f_k(a_k, C_k) \right]. \tag{50}$$

Second,

b) For every $j \in \{1, \ldots, K\}$ we have

$$\mathbb{E}[Y \mid a_1, \ldots, \mathrm{do}(a_j), \ldots, a_K] = \int y \, \mathrm{p}^{\mathcal{M}(\mathrm{do}(a_j))}(y \mid a_1, \ldots, \mathrm{do}(a_j), \ldots, a_K) \, \mathrm{d}y \tag{51}$$

$$= \int \ldots \int y \, \mathrm{p}^{\mathcal{M}(\mathrm{do}(a_j))}(y \mid a_1, \ldots, \mathrm{do}(a_j), \ldots, a_K, c_1, \ldots, c_K) \, \mathrm{d}y$$
$$\times \mathrm{p}^{\mathcal{M}(\mathrm{do}(a_j))}(c_1, \ldots c_K \mid a_1, \ldots, \mathrm{do}(a_j), \ldots, a_K) \, \mathrm{d}c_1 \ldots \mathrm{d}c_K \tag{52}$$

$$= \int \ldots \int y \, \mathrm{p}(y \mid a_1, \ldots, a_j, \ldots, a_K, c_1, \ldots, c_K) \, \mathrm{d}y$$
$$\times \mathrm{p}^{\mathcal{M}(\mathrm{do}(a_j))}(c_1, \ldots c_K \mid a_1, \ldots, \mathrm{do}(a_j), \ldots, a_K) \, \mathrm{d}c_1 \ldots \mathrm{d}c_K \tag{53}$$

$$= \int \ldots \int \mathbb{E}[Y \mid a_1, \ldots, a_j, \ldots, a_K, c_1, \ldots, c_K]$$
$$\times \mathrm{p}^{\mathcal{M}(\mathrm{do}(a_j))}(c_1, \ldots c_K \mid a_1, \ldots, \mathrm{do}(a_j), \ldots, a_K) \, \mathrm{d}c_1 \ldots \mathrm{d}c_K \tag{54}$$

$$= \int \ldots \int \left( \sum_k f_k(a_k, c_k) \right) \mathrm{p}^{\mathcal{M}(\mathrm{do}(a_j))}(c_1, \ldots c_K \mid a_1, \ldots, \mathrm{do}(a_j), \ldots, a_K) \, \mathrm{d}c_1 \ldots \mathrm{d}c_K \tag{55}$$

$$\overset{\text{Lemma 2b)}}{=} \int \ldots \int \left( \sum_k f_k(a_k, c_k) \right) \mathrm{p}(c_j) \prod_{k \neq j} \mathrm{p}(c_k \mid a_1, \ldots, a_K) \, \mathrm{d}c_1 \ldots \mathrm{d}c_K \tag{56}$$

$$= \int f_j(a_j, c_j) \mathrm{p}(c_j) \, \mathrm{d}c_j + \sum_{k \neq j} \int f_k(a_k, c_k) \mathrm{p}(c_k \mid a_1, \ldots, a_K) \, \mathrm{d}c_k \tag{57}$$

$$= \mathbb{E}_{C_j \sim \mathrm{p}(C_j)}[f_j(a_j, C_j)] + \sum_{k \neq j} \mathbb{E}_{C_k \sim \mathrm{p}(C_k \mid a_1, ., a_K)}[f_k(a_k, C_k)]. \tag{58}$$

And finally,

c)

$$\mathbb{E}[Y \mid a_1, \ldots, a_K] = \int y \, \mathrm{p}(y \mid a_1, \ldots, a_K) \, \mathrm{d}y \tag{59}$$

$$= \int \ldots \int y \, \mathrm{p}(y \mid a_1, \ldots, a_K, c_1, \ldots, c_K) \, \mathrm{d}y \, \mathrm{p}(c_1, \ldots c_K \mid a_1, \ldots, a_K) \, \mathrm{d}c_1 \ldots \mathrm{d}c_K \tag{60}$$

$$= \int \ldots \int \mathbb{E}[Y \mid a_1, \ldots, a_K, c_1, \ldots, c_K] \, \mathrm{p}(c_1, \ldots c_K \mid a_1, \ldots, a_K) \, \mathrm{d}c_1 \ldots \mathrm{d}c_K \tag{61}$$

$$= \int \ldots \int \left( \sum_k f_k(a_k, c_k) \right) \mathrm{p}(c_1, \ldots c_K \mid a_1, \ldots, a_K) \, \mathrm{d}c_1 \ldots \mathrm{d}c_K \tag{62}$$

$$\overset{\text{Lemma 2c)}}{=} \int \ldots \int \left( \sum_k f_k(a_k, c_k) \right) \prod_k \mathrm{p}(c_k \mid a_1, \ldots, a_K) \, \mathrm{d}c_1 \ldots \mathrm{d}c_K \tag{63}$$

$$= \sum_k \int f_k(a_k, c_k) \, \mathrm{p}(c_k \mid a_1, \ldots, a_K) \, \mathrm{d}c_k \tag{64}$$

$$= \sum_k \mathbb{E}_{C_k \sim \mathrm{p}(C_k|a_1,.,a_K)}[f_k(a_k, C_k)]. \tag{65}$$

We learn $K$ functions

$$\hat{f}_k(a_1, \dots, a_K, R_k), \qquad k \in \{1, \dots, K\} \tag{66}$$

where $R_k \in \{0, 1\}$ is an indicator for whether $A_k$ was intervened on. $R_k$ can be thought of as selecting one of two functions

$$\hat{f}_k(a_1, \dots, a_K, R_k) = \begin{cases} \hat{f}_k^{\mathrm{obs}}(a_1, \dots, a_K) & \text{if } R_k = 0 \\ \hat{f}_k^{\mathrm{int}}(a_1, \dots, a_K) & \text{if } R_k = 1 \end{cases} \tag{67}$$

where $\hat{f}_k^{\mathrm{obs}}$ and $\hat{f}_k^{\mathrm{int}}$ are universal function approximators.

We define

$$\hat{f}(a_1, \dots, a_K, R_1, \dots, R_K) = \sum_{k=1}^{K} \hat{f}_k(a_1, \dots, a_K, R_k). \tag{68}$$

Since we are in the infinite data regime and have universal function approximators we can fit $\hat{f}$ such that

$$\hat{f}(a_1, \dots, a_K, R_1{=}0, \dots, R_K{=}0) = \mathbb{E}[Y \mid a_1, \dots, a_K] \tag{69}$$

$$\hat{f}(a_1, \dots, a_K, R_1{=}0, \dots, R_j{=}1, \dots, R_K{=}0) = \mathbb{E}[Y \mid a_1, \dots, \mathrm{do}(a_j), \dots, a_K], \quad \text{for } j \in \{1, \dots, K\}. \tag{70}$$

From the definition of Equation (68), we have for each $j \in \{1, \dots, K\}$

$$\hat{f}(a_1, \dots, a_K, R_1{=}0, \dots, R_j{=}1, \dots, R_K{=}0) - \hat{f}(a_1, \dots, a_K, R_1{=}0, \dots, R_j{=}0, \dots, R_K{=}0) \tag{71}$$

$$= \hat{f}_j(a_1, \dots, a_K, R_j{=}1) - \hat{f}_j(a_1, \dots, a_K, R_j{=}0). \tag{72}$$

After training the estimator we have

$$\hat{f}(a_1, \dots, a_K, R_1{=}0, \dots, R_j{=}1, \dots, R_K{=}0) - \hat{f}(a_1, \dots, a_K, R_1{=}0, \dots, R_j{=}0, \dots, R_K{=}0) \tag{73}$$

$$= \mathbb{E}[Y \mid a_1, \dots, \mathrm{do}(a_j), \dots, a_K] - \mathbb{E}[Y \mid a_1, \dots, a_K] \tag{74}$$

$$\overset{(58),(65)}{=} \mathbb{E}_{C_j \sim \mathrm{p}(C_j)}[f_j(a_j, C_j)] - \mathbb{E}_{C_j \sim \mathrm{p}(C_j|a_1,.,a_K)}[f_j(a_j, C_j)] \tag{75}$$

where in the last step we have plugged in the decomposition of the expectation in the single- intervention (58) and observational (65) setting.

Combining the definition of the estimator (68), and Equations (72) and (75), we get an expression for the joint interventional effect:

$$\hat{f}(a_1, \dots, a_K, R_1{=}1, \dots, R_K{=}1) = \sum_{j=1}^{K} \hat{f}_j(a_1, \dots, a_K, R_j{=}1) \tag{76}$$

$$= \sum_{j=1}^{K} \left( \mathbb{E}_{C_j \sim \mathrm{p}(C_j)}[f_j(a_j, C_j)] - \mathbb{E}_{C_j \sim \mathrm{p}(C_j|a_1,.,a_K)}[f_j(a_j, C_j)] + \hat{f}_j(a_1, \dots, a_K, R_j{=}0) \right) \tag{77}$$

$$= \sum_{j=1}^{K} \mathbb{E}_{C_j \sim \mathrm{p}(C_j)}[f_j(a_j, C_j)] - \sum_{j=1}^{K} \mathbb{E}_{C_j \sim \mathrm{p}(C_j|a_1,.,a_K)}[f_j(a_j, C_j)] + \sum_{j=1}^{K} \hat{f}_j(a_1, \dots, a_K, R_j{=}0) \tag{78}$$

$$\overset{(50),(65),(68)}{=} \mathbb{E}[Y \mid \mathrm{do}(a_1, \dots, a_K)] \underbrace{-\mathbb{E}[Y \mid a_1, \dots, a_K] + \hat{f}(a_1, \dots, a_K, R_1{=}0, \dots, R_K{=}0)}_{\overset{(69)}{=}0} \tag{79}$$

$$= \mathbb{E}[Y \mid \mathrm{do}(a_1, \dots, a_K)]. \tag{80}$$

$\blacksquare$

## B. Proof of Proposition 1

**Proof** (Proposition 1) As in the proof of Theorem 1, we can decompose the interventional effect as

$$\mathbb{E}[Y \mid \mathrm{do}(\mathbf{a}_{\mathrm{int}}), \mathbf{a}_{\mathrm{obs}}] = \sum_{\substack{j \\ A_j \in \mathbf{A}_{\mathrm{int}}}} \mathbb{E}_{C_j \sim \mathrm{p}(C_j)}[f_j(a_j, C_j)] + \sum_{\substack{k \\ A_k \in \mathbf{A}_{\mathrm{obs}}}} \mathbb{E}_{C_k \sim \mathrm{p}(C_k \mid a_1,.,a_K)}[f_k(a_k, C_k)]. \tag{81}$$

We learn $K$ functions and fit them to the observational and single-interventional expectation (Equations (66) to (70)).

Let

$$R_k = \begin{cases} 1 \text{ if } A_k \in \mathbf{A}_{\mathrm{int}} \\ 0 \text{ if } A_k \in \mathbf{A}_{\mathrm{obs}} \end{cases} \quad \text{for all } k \in \{1, \dots, K\}. \tag{82}$$

Then our estimator identifies the interventional effect (21), since

$$\hat{f}(a_1, \dots, a_K, R_1, \dots, R_K) \tag{83}$$

$$\stackrel{(67),(68)}{=} \sum_{\substack{k \\ A_k \in \mathbf{A}_{\mathrm{obs}}}} \hat{f}_k(a_1, \dots, a_K, R_k{=}0) + \sum_{\substack{j \\ A_j \in \mathbf{A}_{\mathrm{int}}}} \hat{f}_j(a_1, \dots, a_K, R_j{=}1) \tag{84}$$

$$= \sum_{\substack{k \\ A_k \in \mathbf{A}_{\mathrm{obs}}}} \hat{f}_k(a_1, \dots, a_K, R_k{=}0) + \sum_{\substack{l \\ A_l \in \mathbf{A}_{\mathrm{int}}}} \hat{f}_l(a_1, \dots, a_K, R_l{=}0)$$

$$- \sum_{\substack{l \\ A_l \in \mathbf{A}_{\mathrm{int}}}} \hat{f}_l(a_1, \dots, a_K, R_l{=}0) + \sum_{\substack{j \\ A_j \in \mathbf{A}_{\mathrm{int}}}} \hat{f}_j(a_1, \dots, a_K, R_j{=}1) \tag{85}$$

$$= \underbrace{\sum_{k=1}^{K} \hat{f}_k(a_1, \dots, a_K, R_k{=}0)}_{\stackrel{(68),(69)}{=}\mathbb{E}[Y \mid a_1,.,a_K]} + \sum_{j,\ A_j \in \mathbf{A}_{\mathrm{int}}} \underbrace{\left( \hat{f}_j(a_1, \dots, a_K, R_j{=}1) - \hat{f}_j(a_1, \dots, a_K, R_j{=}0) \right)}_{\stackrel{(72),(75)}{=}\mathbb{E}_{C_j \sim \mathrm{p}(C_j)}[f_j(a_j,C_j)] - \mathbb{E}_{C_j \sim \mathrm{p}(C_j \mid a_1,.,a_K)}[f_j(a_j,C_j)]} \tag{86}$$

$$\stackrel{(65)}{=} \sum_{k=1}^{K} \mathbb{E}_{C_k \sim \mathrm{p}(C_k \mid a_1,.,a_K)}[f_k(a_k, C_k)]$$

$$+ \sum_{j,\ A_j \in \mathbf{A}_{\mathrm{int}}} \left( \mathbb{E}_{C_j \sim \mathrm{p}(C_j)}[f_j(a_j, C_j)] - \mathbb{E}_{C_j \sim \mathrm{p}(C_j \mid a_1,.,a_K)}[f_j(a_j, C_j)] \right) \tag{87}$$

$$= \sum_{\substack{j \\ A_j \in \mathbf{A}_{\mathrm{int}}}} \mathbb{E}_{C_j \sim \mathrm{p}(C_j)}[f_j(a_j, C_j)] + \sum_{\substack{k \\ A_k \in \mathbf{A}_{\mathrm{obs}}}} \mathbb{E}_{C_k \sim \mathrm{p}(C_k \mid a_1,.,a_K)}[f_k(a_k, C_k)] \tag{88}$$

$$\stackrel{(81)}{=} \mathbb{E}[Y \mid \mathrm{do}(\mathbf{a}_{\mathrm{int}}), \mathbf{a}_{\mathrm{obs}}] \tag{89}$$

$$\blacksquare$$

## C. Proof of Corollary 1

**Proof** (Corollary 1) Similar to the proof of Theorem 1, we can decompose the outcome expectations in the joint interventional regime and the settings for which we have data as

$$\mathbb{E}[Y \mid \mathrm{do}(a_1, \dots, a_K)] = \sum_{B \in \mathfrak{B}} \mathbb{E}_{\mathbf{C}_B \sim \prod_{k \in B} \mathrm{p}(C_k)} \left[ f_B(\mathbf{a}_B, \mathbf{C}_B) \right] \tag{90}$$

$$\mathbb{E}[Y \mid \mathrm{do}(\mathbf{a}_B), \mathbf{a}_{\neg B}] = \mathbb{E}_{\mathbf{C}_B \sim \prod_{k \in B} \mathrm{p}(C_k)} \left[ f_B(\mathbf{a}_B, \mathbf{C}_B) \right]$$

$$+ \sum_{\tilde{B} \neq B} \mathbb{E}_{\mathbf{C}_{\tilde{B}} \sim \prod_{k \in \tilde{B}} \mathrm{p}(C_k \mid a_1,.,a_K)} [f_{\tilde{B}}(\mathbf{a}_{\tilde{B}}, \mathbf{C}_{\tilde{B}})] \quad \forall B \in \mathfrak{B} \tag{91}$$

$$\mathbb{E}[Y \mid a_1, \dots, a_K] = \sum_{B \in \mathfrak{B}} \mathbb{E}_{\mathbf{C}_B \sim \prod_{k \in \tilde{B}} \mathrm{p}(C_k \mid a_1,.,a_K)} [f_B(\mathbf{a}_B, \mathbf{C}_B)], \tag{92}$$

where $\mathbf{a}_{\neg B}$ denotes all actions that are not in subset $B$.

We define $|\mathfrak{B}|$ estimator functions as

$$\sum_{B \in \mathfrak{B}} \hat{f}_B(a_1, \dots, a_K, R_B), \tag{93}$$

where $R_B \in \{0, 1\}$ indicates whether the actions in the subset $B$ were intervened on. Then, following the analogous steps as in Theorem 1, the joint interventional effect (6) is identified through

$$\sum_{B \in \mathfrak{B}} \hat{f}_B(a_1, \dots, a_K, R_B{=}1). \tag{94}$$

∎

# D. Probability Distributions for Example 1

Table 1. Observational distribution: $\mathrm{P}^{\mathcal{M}}(Y, A_1, A_2) = \mathrm{P}^{\widetilde{\mathcal{M}}}(Y, A_1, A_2)$

| $\mathrm{P}^{\mathcal{M}}(Y, A_1, A_2)$ | $Y = 0$ | $Y = 1$ |
|---|---|---|
| $A_1 = 0, A_2 = 0$ | $(1 - p)$ | $0$ |
| $A_1 = 1, A_2 = 0$ | $p(1 - p)$ | $0$ |
| $A_1 = 0, A_2 = 1$ | $0$ | $0$ |
| $A_1 = 1, A_2 = 1$ | $p^2(1 - p)$ | $p^3$ |

Table 2. Single-intervention distribution: $\mathrm{P}^{\mathcal{M}(\mathrm{do}(A_1))}(Y, A_2) = \mathrm{P}^{\widetilde{\mathcal{M}}(\mathrm{do}(A_1))}(Y, A_2)$

| | $\mathrm{P}^{\mathcal{M}(\mathrm{do}(A_1))}(Y, A_2)$ | $Y = 0$ | $Y = 1$ |
|---|---|---|---|
| $\mathrm{do}(A_1 = 0)$ | $A_2 = 0$ | $1$ | $0$ |
| | $A_2 = 1$ | $0$ | $0$ |
| $\mathrm{do}(A_1 = 1)$ | $A_2 = 0$ | $(1 - p)$ | $0$ |
| | $A_2 = 1$ | $0$ | $p$ |

Table 3. Single-intervention distribution: $\mathrm{P}^{\mathcal{M}(\mathrm{do}(A_2))}(Y, A_1) = \mathrm{P}^{\widetilde{\mathcal{M}}(\mathrm{do}(A_2))}(Y, A_1)$

| | $\mathrm{P}^{\mathcal{M}(\mathrm{do}(A_2))}(Y, A_1)$ | $Y = 0$ | $Y = 1$ |
|---|---|---|---|
| $\mathrm{do}(A_2 = 0)$ | $A_1 = 0$ | $(1 - p)$ | $0$ |
| | $A_1 = 1$ | $p(1 - p)$ | $p^2$ |
| $\mathrm{do}(A_2 = 1)$ | $A_1 = 0$ | $(1 - p)$ | $0$ |
| | $A_1 = 1$ | $p(1 - p)$ | $p^2$ |

# E. Experiments

Table 4. Standard deviations of the exogenous noise variables in the synthetic experiments.

| Exogenous noise variable | std |
|---|---|
| $U$ | 0.1 |
| $V_1, \dots, V_5$ | 0.1 |
| $W_1, \dots, W_5$ | 0.5 |

**Sampling SCMs.** We sample SCMs of the form

$$Y := \sum_{k=1}^{5} f_k(A_k, C_k) + U \tag{95}$$

$$A_k := g_k(C_k, \mathbf{A}_{\text{pa}(k)}) \qquad \text{for } k \in 1, \dots, 5 \tag{96}$$

$$C_k := W_k \qquad \text{for } k \in 1, \dots, 5 \tag{97}$$

where the functions $f_k, g_k$ are third order polynomials, and $\mathbf{A}_{\text{pa}(k)}$ are the actions that are also parents of action $A_k$. Each potential edge $A_j \to A_k$ with $k > j$ is drawn at random from Bernoulli($p_{\text{edge}}$), where the edge probability $p_{\text{edge}}$ is set to 0.3 in our experiments. The distributions of the exogenous noise variables $\{U, V_1, \dots, V_5, W_1, \dots, W_5\}$ have zero mean are either Gaussian, Uniform or Logistic, which is chosen at random. The standard deviations are given in Table 4.

The polynomial functions $f_k$ and $g_k$ are also randomly generated. For each function, we create a multivariate polynomial of second order with mixed terms. The process for generating these functions is as follows:

1. We consider all possible combinations of powers for the input variables, up to the second order. For a function with $n$ input variables, we consider all non-negative integer power combinations $(p_1, \dots, p_n)$ such that $\sum_{i=1}^{n} p_i \leq 2$.

2. For each of these power combinations, we generate a random coefficient drawn from a normal distribution with mean 0 and a small standard deviation $\sigma$ (in our case, $\sigma = 0.1$).

3. To ensure that the scales of the variables do not grow exponentially along the topological order, we normalize the coefficients. This is done by dividing each coefficient by the sum of the absolute values of all coefficients.

The resulting polynomial function for each $f_k$ takes the form:

$$f_k(A_k, C_k) = \sum_{i,j} \alpha_{ij} A_k^i C_k^j \tag{98}$$

where $i + j \leq 2$ (since we're using second-order polynomials), and $\sum_{i,j} |\alpha_{ij}| = 1$ due to the normalization. Similarly, each $g_k$ is a polynomial function of $C_k$ and the parent actions $\mathbf{A}_{\text{pa}(k)}$, with the same properties of being second-order and having normalized coefficients. For a $g_k$ with $m$ parent actions, the function takes the form:

$$g_k(C_k, A_{p_1}, \dots, A_{p_m}) = \sum_{i_0, i_1, \dots, i_m} \alpha_{i_0 i_1 \dots i_m} C_k^{i_0} A_{p_1}^{i_1} \cdots A_{p_m}^{i_m} \tag{99}$$

where $\sum_{j=0}^{m} i_j \leq 2$ and $\sum_{i_0, i_1, \dots, i_m} |\alpha_{i_0 i_1 \dots i_m}| = 1$.

In order to satisfy Assumption 1 on the support of the interventions, we sample single interventions and joint interventions on the action variables to match the observational distributions. That is, we sample intervention values from a normal distribution:

$$A_k^{\text{int}} \sim \mathcal{N}(\hat{\mu}_k, \hat{\sigma}_k^2) \tag{100}$$

where $\hat{\mu}_k$ and $\hat{\sigma}_k$ are the empirical mean and standard deviation of $A_k$ in the observational data, respectively. For joint interventions, the intervention values are sampled independently for each action variable, following the same distribution as in the single intervention case.

**Estimators.** For each interventional setting

$$\mathbb{E}[Y \mid a_1, \dots, a_K] \tag{101}$$

$$\mathbb{E}[Y \mid a_1, \dots, \text{do}(a_j), \dots, a_K], \quad \text{for } j \in \{1, \dots, K\} \tag{102}$$

we fit a third order polynomial estimator. Hence, the estimators

$$\hat{f}(a_1, \dots, a_K, R_1{=}0, \dots, R_K{=}0) \tag{103}$$

$$\hat{f}(a_1, \dots, a_K, R_1{=}0, \dots, R_j{=}1, \dots, R_K{=}0), \quad \text{for } j \in \{1, \dots, K\}, \tag{104}$$

are represented by $K + 1$ polynomials. Hence, the estimator functions for the effect of each action $\hat{f}_k$ are represented implicitly and can be recovered by adding and subtracting the corresponding terms. For example,

$$\hat{f}_j(a_1, \dots, a_K, R_k{=}0) = \hat{f}(a_1, \dots, a_K, R_1{=}0, \dots, R_j{=}1, \dots, R_K{=}0) - \hat{f}(a_1, \dots, a_K, R_1{=}0, \dots, R_K{=}0) \,. \quad (105)$$

We regularize the estimators using Ridge regression and find the optimal regularization parameter through 3-fold cross validation for each estimator.

**Evaluation.**  We evaluate all models on the test dataset of the joint interventional data. We report average root mean squared errors and the corresponding standard error of the mean.

**Experimental settings.**

- In Figure 2(a), we generate $N_{\text{obs}} = N_{\text{sint}} = N_{\text{obs}} = 10^6$ samples for the observational, 5 single interventional and the joint interventional dataset. For the observational baseline, we train one estimator on the observational samples. For the joint interventional baseline, we train one estimator on the joint interventional samples. Then we train one estimator on pooled observational and single-interventional data.

- Figure 2(b) shows the prediction error for varying ratios of single-interventional and observational samples. The total number of data points is kept constant at $N_{\text{tot}} = (K + 1) \times 10^4$ samples. The observational and joint interventional baselines represent estimators trained on $N_{\text{tot}}$ samples from their respective datasets.

- For each point in Figure 2(c) the Intervention Generalization method is trained on $N_{\text{tot}} = N_{\text{obs}} + KN_{\text{sint}}$ data points, where the observational and single-interventional data sets have the same number of samples, that is, $N_{\text{obs}} = N_{\text{sint}}$. We compare its prediction error for the joint interventional effect to an estimator trained on $N_{\text{tot}}$ joint interventional samples.

### E.1. Additional Results

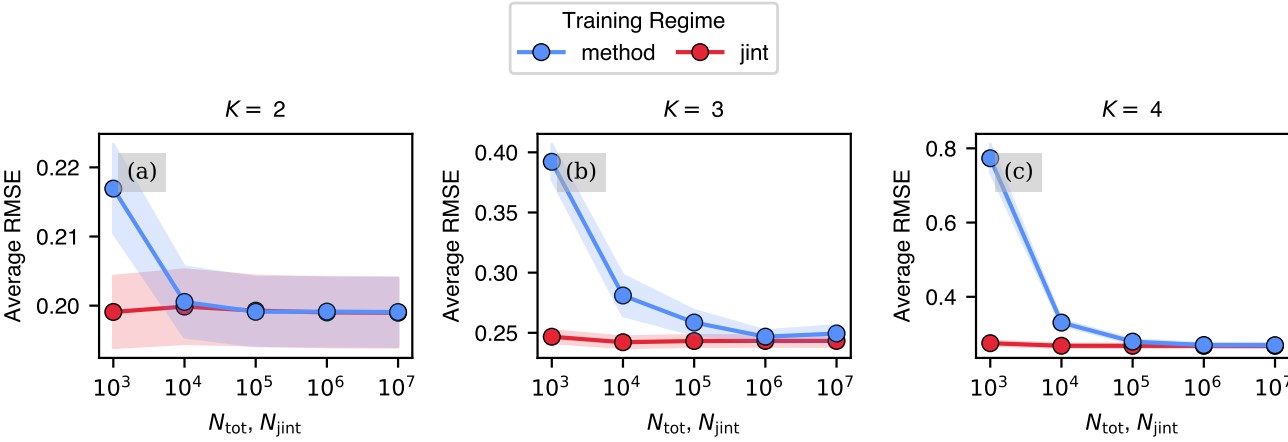

*Figure A.* **Convergence of the Intervention Generalization Method.** Average root mean squared error (RMSE) for predicting the joint interventional effect $\mathbb{E}[Y \mid \mathrm{do}(a_1, \dots, a_5)]$ for different numbers of total data points $N_{\text{tot}}$. Each data point in the plot is averaged over 100 randomly sampled SCMs. The columns show different numbers of action variables $K$.

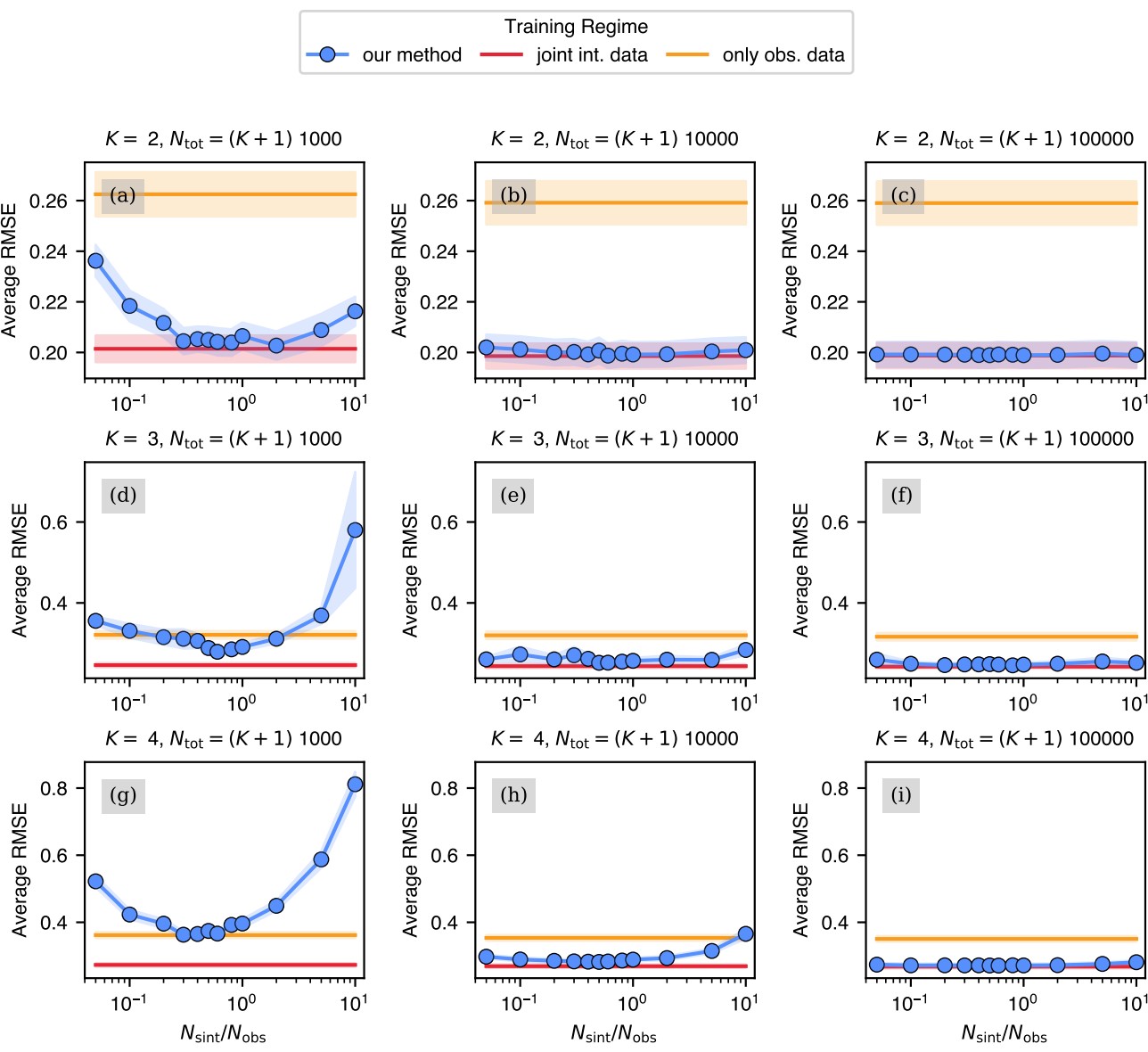

*Figure B.* **Experiments with Varying Ratios of Single-Interventional and Observational Data.** Average root mean squared error (RMSE) for predicting the joint interventional effect $\mathbb{E}[Y \mid \mathrm{do}(a_1, \dots, a_5)]$ for different ratios $N_{\mathrm{sint}}/N_{\mathrm{obs}}$, while keeping the total number of data points $N_{\mathrm{tot}} = N_{\mathrm{obs}} + K N_{\mathrm{sint}}$ constant. Each data point in the plot is averaged over 100 randomly sampled SCMs. The observational and joint interventional baselines represent estimators trained on $N_{\mathrm{tot}}$ samples from their respective datasets. The columns correspond to different total numbers of data points $N_{\mathrm{tot}}$, while the rows show different numbers of action variables $K$.

