# OpenReview forum: "Learning Joint Interventional Effects from Single-Variable Interventions in Additive Models"
_ICML.cc/2025/Conference — ICML 2025 poster_

### Official Review · Reviewer_2XuD · 2025-03-06

**Overall Recommendation:** 4

**Summary:**

The manuscript studies the problem of intervention generalization in additive causal models. Specifically, the authors consider a data generating process in which an outcome variable $Y$ is an additive function of actions $A_1, \dots, A_K$ and unobserved confounders $C_1, \dots, C_K$. Given some set of observational data and single-intervention experiments, the method estimates a joint intervention effect $\mathbb E[Y \mid do(a_1, \dots, a_K)]$. Several identifiability results are presented, along with experiments illustrating the utility of their proposal.

## update after rebuttal

I thank the authors for their reply and look forward to seeing the extended benchmarks should the paper be accepted. I will maintain my score of 4.

**Claims And Evidence:**

The paper is primarily theoretical, with the joint and mixed identifiability results taking center stage. The proof sketch helps walk the reader through the authors' reasoning on this point. The experiments are a little perfunctory (more on this below), but sufficient to illustrate the principle.

**Essential References Not Discussed:**

See the aforementioned works from Caroline Uhler's lab. These may not be identical to the problem setting here, but they're certainly in the neighborhood.

**Experimental Designs Or Analyses:**

As noted above, the experimental DGP is sound but comparisons are a little thin.

**Methods And Evaluation Criteria:**

The data generating process for the experiments makes good sense, but the alternative estimators seem a little naive. Several intervention generalization papers were mentioned in the Related Work section, and I was expecting to see these return in the benchmarks. (I'm thinking specifically of Bravo-Hermsdorff et al., Jung et al., and Saengkyongam & Silva.) Are these methods inapplicable in this setting? If so, why? Caroline Uhler's lab has also done considerable work in this area but I don't believe any of those methods are cited (see below).

-https://arxiv.org/abs/2405.19225

-https://proceedings.mlr.press/v236/ribot24a.html

**Other Comments Or Suggestions:**

N/A

**Other Strengths And Weaknesses:**

The paper is clear and convincing. I think with just a couple of minor amendments – namely, grounding the discussion with a running example and extending the benchmarks to include alternative intervention generalization techniques – it could realize its full potential.

**Questions For Authors:**

See comments above.

**Relation To Broader Scientific Literature:**

This work makes a meaningful contribution to the intervention generalization literature, which has numerous applications in science and industry. One recommendation to help improve the manuscript – there are several potential use cases cited in the introduction, but these are basically never mentioned again after the first paragraph. Readers may benefit from a running example that grounds us as we move through theoretical and experimental results.

**Theoretical Claims:**

Theoretical results are exceptionally clear and well presented.

---

> ### Author Rebuttal · Authors · 2025-04-01
>
> # Response to Reviewer 2XuD
>
> We thank the reviewer for the valuable comments and the pointers to Caroline Uhler’s lab related research. We would like to address some of the reviewer’s concerns.
>
> On the baselines: we agree with the reviewer that using the purely observational data the pooled data might not be a fair comparison due to their simplicity. However, using joint interventional data is the closest one gets to an “oracle” version, since then the problem is reduced to estimating a conditional function $P(Y \mid do(A))$. Moreover we believe it would be unfair to use the methods of Bravo-Hermsdorff et.al., Jung et.al. and Saengkyongam & Silva as benchmarks, as their assumptions usually do not match ours.
>
> In particular, we cannot apply Saengkyongam & Silva’s method because their method requires the noise to be Gaussian (indeed, our method is also applicable for discrete variables). Likewise, our setting with the additive outcome mechanism is not identifiable in the interventional factor graph model by Bravo-Hermsdorff, since the additive structure of the causal mechanism does not translate to an extra factorization of the joint probability in the different interventional settings.
>
> About Caroline Uhler’s lab related research: We thank the reviewer for the pointers. These are indeed quite related to our work.
>
> - In particular Ribot et.al. is also interested in causal effect estimation in novel scenarios when some action-context pairs have been observed, which in the language of our paper would amount to estimating joint causal effects from observing only certain subsets of the interventions. Likewise, their identification strategy relies on making a linearity assumption, which shows that a latent factor model is induced.
> - On the other hand, Bijan et.al. cluster variables with respect to interventions that come from a mixture distribution and then use the clustering to produce Synthetic Potential Outcomes to compute a previously unseen intervention. Their setting differ from ours in that we have several possible actions as opposed to a single treatment and we make an assumption on the functional form of the SCM in order to make the joint interventions identifiable.
>
> We will add an extended comparison in the camera-ready version of the paper.
>
> On the running example: We believe that having a running example is a good idea. However, there are two reasons why we don’t include it. First, lack of space in the initial submission, and second, including a running example on the camera-ready version would potentially require large changes to the paper, which would be challenging to do given the time constraints and difficult to review.
>
> We appreciate the time and effort you have invested in providing constructive feedback, which will help us improve our paper.

---

### Official Review · Reviewer_tbRD · 2025-03-07

**Overall Recommendation:** 2

**Summary:**

This paper presents a novel construction for identifiability of joint interventional effect from single variable intervention results, with additive assumptions placed on the causal mechanism. The method is tested on synthetic data and demonstrated effectiveness and on pare performance when compare with a model trained on joint interventional data.

**Claims And Evidence:**

The claims are in general correct with enough evidence.

**Essential References Not Discussed:**

I think most of the relevant literatures are discussed in the paper.

**Experimental Designs Or Analyses:**

The experimental design and analysis seems to be fine, although (1) semi-synthetic experiment could be used for better representing its real-world effect (2) better motivation need for assumption 2 (Additive Outcome Mechanism) and (3) discussion should be made in case of violation of assumption 2 (e.g. feedback loop or no direct ordering in DAG as in "Sachs, K., Perez, O., Pe'er, D., Lauffenburger, D.A. and Nolan, G.P., 2005. Causal protein-signaling networks derived from multiparameter single-cell data. Science, 308(5721), pp.523-529.") See more in question section.

**Methods And Evaluation Criteria:**

The proposed methods are correct and evaluation seems to be reasonable.

**Other Comments Or Suggestions:**

Please see my comments earlier.


## update after rebuttal

I thank the authors for the detailed and thoughtful responses. While the clarifications are helpful and address some of my concerns, I remain unconvinced about the real-world impact (in terms of application) of this research and empirical strength of the paper (with synthetic data only) under the current assumptions. Therefore, my score remains unchanged.

**Other Strengths And Weaknesses:**

The paper is well-motivated and provide clear contributions with experimental validations, the theory is in general correct but I am a bit less convinced its impact on real-world scenarios as (1) the additive assumption is nothing new and not very novel (one can claim this is novel as it is not placed on the mechanism or functional forms instead of parameters but sill this is not a major novel contribution) (2) the limitation of having confounder $C_{i}$ point to $X_{i}$ only seems to be restrictive and might not be applicable in real-world (how about have $C_{i}$ point to $X_{i}$ and $X_{j}$?), please refers to comments in "**Experimental Designs Or Analyses**" section. (3) experimental results are quite weak as it use purely synthetic data without constraining on its parameters (100 different SCMs helps but still quite weak), my suggestion is to a.) learn parameters from real-world data and b.)have discussion on when the identifiable results does not apply and conduct experiments under this to give us a better understanding on how applicable the results are in more realistic setting.

**Questions For Authors:**

I still found the justification is weak for why the authors focus on parametric identification this case? The applicable of this method for real-world problem can be completely trashed if the data-generation process is not under this constrained form.

Could you please answer some of my concerns in the comments above?

**Relation To Broader Scientific Literature:**

This is an interesting and one of the frontier questions in the causal inference field, learning the joint effect from single causal effects. The key ingredient of this paper is the additive causal mechanism assumption, which is interesting but also quite strong and restrictive to real world scenarios.

**Theoretical Claims:**

Yes. the identifiable results seems to be correct.

---

> ### Author Rebuttal · Authors · 2025-04-01
>
> Thank you for taking the time to review our paper and for your thoughtful comments. We appreciate your feedback and would like to address your concerns below.
>
> ## On the need for parametric identification
>
> From g-ID theory [1], we know that our problem setting is non-identifiable in the non-parametric case (i.e., only considering the causal structure in Fig. 1). I.e., given single-interventional and observational data, the joint-interventional effect cannot be uniquely determined in general. The causal structure, together with the intervention sets, forms what's known as a "thicket" (Def. 6 in [1]), which precludes identifiability. We believe that such multi-cause settings are interesting across many domains (see citations in Introduction). Unfortunately, these settings are limited by non-identifiability in the intervention generalization setting without additional assumptions.
>
> ## On the novelty of the additivity assumption
>
> We don't claim the additivity assumption itself is novel. There is indeed a rich literature on additivity in statistics [2,3]. In causal inference, additivity of exogenous noise has been quite common. However, functional additivity, where the function is composed of a sum of nonlinear functions, to the best of our knowledge, has not been explored in causal inference. In particular, it has not been investigated as a parametric assumption in the intervention generalization setting.
>
> This function class is particularly interesting here since it limits the complexity of interaction between actions in the outcome mechanism. This effectively prevents two distinct causal models from agreeing across single-interventional regimes while differing on the joint-interventional.
>
> ## On the limitations of Assumption 2
>
> We agree that Ass. 2 limits the generality of our results. However, as discussed above, the problem is hard and not solvable in the most general case. For many if not most real-world settings with confounding between outcome and actions, intervention generalization is impossible without additional assumptions.
>
> The key question becomes: "What parametric and structural assumptions do we have to make to achieve identifiability in the intervention generalization setting?" In our work, we have shown that under our set of assumptions using the additive outcome mechanism, intervention generalization is possible, and we have provided a practical estimator for it.
>
> ## On the practical applicability
>
> While intervention generalization may not be achievable in all real-world cases, the benefits for the cases where it is applicable are immense. We can reduce the required experimental conditions from growing exponentially with the No. of actions to a linear regime. This makes a significant difference in many real-world settings, where obtaining experimental data is often very hard or expensive.
>
> Additionally, Cor. 1 provides a way to trade-off assumptions on additivity for additional experimental data. If for practical applications it is unclear whether a subset of actions contributes additively, we can choose drop the assumption on this subset. This then comes at the cost of having to collect joint interventional data for this subset. We think that allowing for this trade-off between additivity and joint interventional data makes our method applicable to a broader range of real-world cases than full additivity would allow.
>
> ## On future directions
>
> Our set of assumptions may not be the most parsimonious, and we hope that future work will find more general solutions. One motivation for choosing the additive model class was its relation to more general model classes, namely Generalized Additive Models (GAMs) [4], which in turn are related to the general functions of the Kolmogorov-Arnold Theorem [5]. We hope that our work can serve as a stepping stone toward proving identifiability results for the GAM case.
>
> ## On extensions to more general confounding between actions
>
> If we add confounding between actions, Lem. 1 would not hold. However, we believe this is not a complete blocker for the problem. Our identifiability proof is agnostic to the causal structure between the actions (assuming it has a DAG structure). We suppose that addressing more general action confounding would require modeling the dependencies between the actions as well as the outcome mechanism to achieve intervention generalization. We hope to make progress in this direction in future work.
>
> ## References
> [1] Lee, S. et al. "General identifiability with arbitrary surrogate experiments." 2020.
>
> [2] Friedman, J. H. et al. "Projection pursuit regression." 1981.
>
> [3] Breiman, L. et al. "Estimating optimal transformations for multiple regression and correlation." 1985.
>
> [4] Hastie, T. J. et al. "Generalized Additive Models." 1990.
>
> [5] Kolmogorov, A.N. "On the representation of continuous functions of several variables as superpositions of continuous functions of a smaller number of variables." 1956.

---

### Official Review · Reviewer_9nyV · 2025-03-12

**Overall Recommendation:** 3

**Summary:**

The paper proposes a method for estimating joint interventional effects using observational data and single-variable interventions within nonlinear additive models. It establishes identifiability results, showing that joint effects are recoverable without direct joint interventional data. The authors validate their approach empirically using synthetic data. However, there are some concerns about the novelty of the proposed method.

**Claims And Evidence:**

The authors convincingly support their claims regarding identifiability and practical effectiveness through clear theoretical results and robust empirical evidence. No significant problematic claims are evident, as each theoretical assertion is carefully supported by rigorous mathematical justification and experiments.

**Essential References Not Discussed:**

No.

**Experimental Designs Or Analyses:**

The experimental design utilizing synthetic structural causal models (SCMs) is valid and well-executed. The analyses, particularly comparisons with relevant baselines, including observational-only and joint-interventional estimators, are sound and provide strong validation of the proposed approach.

**Methods And Evaluation Criteria:**

The method makes sense, but I think the assumptions and inteventional data are not necessary.

**Other Comments Or Suggestions:**

Fig.1 is too big.

**Other Strengths And Weaknesses:**

I apologize for my oversight in missing the statement that $C$ represents a set of unobserved confounders. I have updated my score accordingly. I suggest that the authors consider using different formats in Fig. 1 to clearly distinguish between latent variables and observable variables.

#----------------------------------#

I think Assumption 1 is sufficient for identifying the joint interventional effect with only observational data in the setting of the paper. According to Def. 5 and Thm. 3 of [1], the joint interventional effect is identifiable by covariate adjustment with adjustment set being {$C_1,\cdots,C_k$}. The reason that the two models in Example 1 are with different causal effects is the violation of Assumption 1. Hence, there is no novelty in the current method, which additionally uses interventional data and Assumption 2. I am not sure that my comments are completely correct. If the authors could correct me, I am happy to increase my score.

[1] On the validity of covariate adjustment for estimating causal effects. UAI2010

**Questions For Authors:**

See Other Strengths And Weaknesses

**Relation To Broader Scientific Literature:**

The problem is vital in the scientific literature.

**Theoretical Claims:**

I did not check the proofs, but I believe they are correct, because it seems that the results have been verified by previous studies.

---

> ### Author Rebuttal · Authors · 2025-04-01
>
> # Response to Reviewer 9nyV
>
> Thank you for your thoughtful review and for engaging with our work. We appreciate your careful reading and feedback.
>
> We would like to address your primary concern regarding the novelty of our approach and the necessity of our assumptions:
> You mentioned that Assumption 1 might be sufficient for identifying the joint interventional effect using only observational data, referring to Definition 5 and Theorem 3 in "On the validity of covariate adjustment for estimating causal effects" (UAI 2010).
>
> **Critical clarification:** In our problem setting, the confounders ${C_1, \ldots, C_K}$ are unobserved. This is a fundamental aspect of our work, explicitly stated in Section 2.2 where we define $C = {C_1, \ldots, C_K}$ as "a set of unobserved confounders."
>
> The covariate adjustment approach you reference would indeed be applicable if these confounders were observed. However, when confounders are unobserved (as in our setting), covariate adjustment is not possible, and identification becomes significantly more challenging. This is precisely why we need both Assumption 1 (Intervention Support) and Assumption 2 (Additive Outcome Mechanism), along with single-variable interventional data, to achieve identification.
>
> We believe this clarification addresses your main concern about the novelty and necessity of our approach. Thank you again for your review, and we hope this clarifies the contribution of our work.

---

> > ### Comment · Reviewer_9nyV · 2025-04-01
> >
> > I apologize for my oversight in missing the statement that
> >  represents a set of unobserved confounders. I have updated my score accordingly. I suggest that the authors consider using different formats in Fig. 1 to clearly distinguish between latent variables and observable variables.

---

### Official Review · Reviewer_dotZ · 2025-03-14

**Overall Recommendation:** 3

**Summary:**

This paper addresses the challenge of unobserved confounding in multi-treatment (joint intervention) causal inference. It introduces a method leveraging observational data and experimental data where interventions are applied to only one variable at a time, under the assumption of a nonlinear additive outcome and single-cause confounding.

## Update after rebuttal:
I thank the authors for their response. While the theoretical results are well presented, the real-world impact is not entirely clear and the untestable assumptions require strong justification. I am maintaining my score of 3.

**Claims And Evidence:**

The paper clearly establishes identifiability under the specified assumptions. Additionally, the assumptions required for identifiability are transparently stated and reasonably justified. Empirically, the authors convincingly demonstrate that their proposed estimator performs close to a model trained directly on joint interventional data as sample size increases. And it significantly outperforms baselines such as observational-only estimation or naive pooling of single-variable interventions.

**Essential References Not Discussed:**

This work is related to multi-treatment causal inference literature. For example:
- [1] Miao et al. (2023) Identifying effects of multiple treatments in the presence of unmeasured confounding
- [2] Wang and Blei (2020) The blessings of multiple causes
- [3] Zheng et al. (2021) Copula-based Sensitivity Analysis for Multi-Treatment Causal Inference with Unobserved Confounding

**Experimental Designs Or Analyses:**

See above.

**Methods And Evaluation Criteria:**

1. The proposed method relies on strong assumptions, particularly an additive outcome mechanism and single-cause (pair-wise) confounding, where each confounder influences exactly one treatment variable and the outcome. Such strict assumptions limit the generalizability and practical applicability of the method, as real-world scenarios often involve shared confounding across multiple treatments  and complex outcome mechanism.
2. The empirical evaluation is limited to relatively simple synthetic experiments, which may not adequately capture the complexities of real-world causal structures or realistic data generation processes. Investigations into robustness against violations of key assumptions can also clarify the practical utility and limitations of the proposed approach.

**Other Comments Or Suggestions:**

It seems Equation (2) is not compatible with the approach and graph.

**Other Strengths And Weaknesses:**

Overall, this paper is well written and clearly structured.

**Questions For Authors:**

None

**Relation To Broader Scientific Literature:**

The paper positions itself at the intersection of causal identifiability theory, intervention generalization, and additive statistical modeling, combining insights from these areas to propose a novel method for estimating joint interventional effects from single-variable interventions.

**Theoretical Claims:**

Look fine to me.

---

> ### Author Rebuttal · Authors · 2025-04-01
>
> # Response to Reviewer dotZ
>
> Thank you for your thoughtful review and constructive feedback on our paper. We appreciate the time you've invested and would like to address your key concerns.
>
> ## On the strong assumptions of our method
>
> We acknowledge that our model makes strong assumptions, particularly regarding the additive outcome mechanism and the single-cause confounding structure. These assumptions indeed limit the generalizability to certain real-world scenarios where shared confounding or more complex outcome mechanisms might be present.
>
> From general identifiability theory [1,2], we know that the intervention generalization problem is non-identifiable in the non-parametric case (considering only the causal structure). Without additional assumptions, given single-interventional and observational data, the joint-interventional effect cannot be uniquely determined in general.
>
> The key question becomes: "What minimal set of assumptions enables identification in this setting?" Our work proposes one such set of assumptions that makes the problem identifiable. We view this as a starting point rather than a complete solution to all real-world scenarios with confounding.
>
> Importantly, our Corollary 1 provides a way to trade-off assumptions on additivity for additional experimental data. If for practical applications it is unclear whether a subset of actions contributes additively, we can choose not to make that assumption on this subset. This then comes at the cost of having to collect joint interventional data for this subset. We think that allowing for this trade-off between additivity and joint interventional data makes our method applicable to a broader range of real-world cases than full additivity would allow.
>
> In many domains where additive models are already commonly used (e.g., marketing analytics [3], certain pharmacological interactions [4]), our approach could provide practical value despite its limitations. We hope future work will find more general solutions with weaker assumptions that cover a broader range of real-world scenarios. (See also our response to Reviewer tbRD.)
>
> ## On the multi-treatment causal inference literature
>
> You make an excellent point about the connection to multi-treatment causal inference literature. We thank you for suggesting these important references. Our work is indeed related to this literature, and we will add a discussion of these connections in our Related Works section. The papers you suggested (Miao et al., 2023; Wang and Blei, 2020; Zheng et al., 2021) provide valuable context for our work and would strengthen our paper's positioning within the broader causal inference literature.
>
> ## On the limitations of synthetic experiments
>
> Our main goal in the experimental section was to show that, while we have shown that the problem setting is identifiable, the joint-interventional effect is practically estimatable from data with our proposed estimator.
>
> We agree that our empirical evaluation using synthetic data has limitations in representing the full complexity of real-world causal structures. While we've tried to incorporate some level of complexity through randomly generated SCMs with varying parameters, this approach cannot fully capture all real-world nuances.
>
> Thank you again for highlighting these important limitations. Your feedback will help us improve our research and better understand the practical applicability of our method.
>
> ## References
>
> [1] Lee, S., Correa, J. D., and Bareinboim, E. "General identifiability with arbitrary surrogate experiments." Uncertainty in artificial intelligence. PMLR, 2020.
>
> [2] Kivva, Y., et al. "Revisiting the general identifiability problem." Uncertainty in Artificial Intelligence. PMLR, 2022.
>
> [3] Chan, D., & Perry, M. "Challenges and opportunities in media mix modeling." Google Inc, 16, 2017.
>
> [4] Pearson, R. A., Wicha, S. G., & Okour, M. "Drug combination modeling: methods and applications in drug development." The Journal of Clinical Pharmacology, 63(2), 151-165, 2023.

---

> > ### Comment · Reviewer_dotZ · 2025-04-06
> >
> > I'm not very concerned about the additivity assumption in this setting, since there is no free lunch. But I'm more interested in what happens if there is shared confounding, which is the main motivation behind many multi-treatment (as mentioned above) and multi-outcome approaches like [1]. Can Corollary 1 be generalized to confounding shared across interventions within the same subset? If not, real-world examples or more justification of the single-cause confounding assumption could help strengthen the paper's practical applicability.
> >
> > [1] Zheng, J., et al. "Sensitivity to unobserved confounding in studies with factor-structured outcomes." Journal of the American Statistical Association 119.547 (2024): 2026-2037.

---

> > > ### Author Response · Authors · 2025-04-07
> > >
> > > # Response to Reviewer dotZ's Follow-up Comment
> > >
> > > Thank you for raising this important point about shared confounding across interventions, which we should have explicitly mentioned in our paper.
> > >
> > > You're absolutely correct, our framework actually allows for shared confounding within action variables in a given partition (Definition 2). Specifically, Corollary 1 allows us to trade off assumptions in two ways:
> > >
> > > 1. We can relax the additivity assumption for certain subsets of action variables in the outcome mechanism (9) by collecting joint interventional data on those subsets.
> > >
> > > 2. Similarly, we can allow for shared confounding among action variables within the same partition subset, provided we collect joint interventional data on that subset.
> > >
> > > We will clarify these points in Section 6.3 of our revised manuscript.
> > >
> > > However, unobserved confounding between different partition subsets would not be covered by our approach, as Lemma 1 would no longer hold. If we introduce confounding between actions across different partitions, the conditional distributions of confounders would change across interventional settings, breaking the decomposition in our proof.
> > >
> > > Addressing more general confounding structures between actions would likely require explicitly modeling the dependencies between the actions alongside the outcome mechanism. While our current proof is agnostic to the causal structure between actions (assuming it has a DAG structure), extending to shared confounding across partitions remains an important direction for future work.
> > >
> > > Thank you again for this insightful feedback that helps strengthen our paper.

---

### Decision · Program_Chairs · 2025-05-01

**Decision:**

Accept (poster)

**Comment:**

This paper studies the problem of estimating joint interventional effects from observational and single-intervention data under a nonlinear additive outcome model. The theoretical results are clearly presented, and the proposed estimator shows promising performance on synthetic data.

Reviewers agree that the problem is important and the paper is well written. However, there are concerns about the strength of the identifiability assumptions and the limited empirical validation. The method relies on an additive outcome mechanism and single-cause confounding, which may limit real-world applicability. Moreover, the evaluation is restricted to synthetic data without comparisons to more recent baselines. Overall, I recommend a Weak Accept.